# Mitochondria: A Promising Convergent Target for the Treatment of Amyotrophic Lateral Sclerosis

**DOI:** 10.3390/cells13030248

**Published:** 2024-01-29

**Authors:** Teresa Cunha-Oliveira, Liliana Montezinho, Rui F. Simões, Marcelo Carvalho, Elisabete Ferreiro, Filomena S. G. Silva

**Affiliations:** 1CNC—Center for Neuroscience and Cell Biology, CIBB—Centre for Innovative Biomedicine and Biotechnology, University of Coimbra, 3004-504 Coimbra, Portugal; 2Center for Investigation Vasco da Gama (CIVG), Escola Universitária Vasco da Gama, 3020-210 Coimbra, Portugal; lilianamontezinho@gmail.com; 3Mitotag Lda, Biocant Park, 3060-197 Cantanhede, Portugal

**Keywords:** amyotrophic lateral sclerosis, mitochondrial dysfunction, neurodegeneration, pathological mechanisms, motor neuron degeneration, mitochondria-targeted therapies, therapeutic interventions

## Abstract

Amyotrophic lateral sclerosis (ALS) is a devastating neurodegenerative disease characterized by the progressive loss of motor neurons, for which current treatment options are limited. Recent studies have shed light on the role of mitochondria in ALS pathogenesis, making them an attractive therapeutic intervention target. This review contains a very comprehensive critical description of the involvement of mitochondria and mitochondria-mediated mechanisms in ALS. The review covers several key areas related to mitochondria in ALS, including impaired mitochondrial function, mitochondrial bioenergetics, reactive oxygen species, metabolic processes and energy metabolism, mitochondrial dynamics, turnover, autophagy and mitophagy, impaired mitochondrial transport, and apoptosis. This review also highlights preclinical and clinical studies that have investigated various mitochondria-targeted therapies for ALS treatment. These include strategies to improve mitochondrial function, such as the use of dichloroacetate, ketogenic and high-fat diets, acetyl-carnitine, and mitochondria-targeted antioxidants. Additionally, antiapoptotic agents, like the mPTP-targeting agents minocycline and rasagiline, are discussed. The paper aims to contribute to the identification of effective mitochondria-targeted therapies for ALS treatment by synthesizing the current understanding of the role of mitochondria in ALS pathogenesis and reviewing potential convergent therapeutic interventions. The complex interplay between mitochondria and the pathogenic mechanisms of ALS holds promise for the development of novel treatment strategies to combat this devastating disease.

## 1. Introduction

Amyotrophic lateral sclerosis (ALS) disease, also known as Lou Gehrig’s disease or Charcot disease, is a progressive neurodegenerative disorder that often manifests quietly, initially concealing its full spectrum of symptoms. The disease advances swiftly, not affecting cognition and self-awareness, yielding a definitive diagnosis that compels patients and their loved ones to confront a daunting reality—a verdict that entails grappling with an immensely debilitating condition, necessitating increasing levels of both human and mechanical support. Regrettably, ALS lacks a cure and frequently culminates in rapid fatality.

Although advances in the understanding of ALS have been made, an effective treatment remains to be discovered. The incidence of ALS ranges from 0.6 to 3.8 per 100,000 person-years, with a prevalence of 4.1 to 8.4 per 100,000 individuals. This predominantly affects individuals between the ages of 51 and 66 [1]. The survival period from the onset of symptoms to demise spans from 24 to 50 months, although a minority—10% of ALS patients—experience a slower variant with survival exceeding a decade [1]. Roughly 90–95% of patients develop sporadic ALS (sALS), characterized by an enigmatic etiology and the absence of familial precedent [2]. The remaining 5–10% manifest familial ALS (fALS) [2], an inherited form, frequently linked to an earlier age of onset [3,4].

Presently, approximately 50 genes have been identified as having a role in disease modification [5], albeit only a fraction of these genes demonstrate a significative prevalence in ALS, particularly within the context of fALS [6]. Among this collection of genes, a subset has shown involvement in both sALS and fALS. Notably, the genes *C9ORF72* (encoding chromosome 9 open reading frame 72 (C9orf72) protein) [7,8], *SOD1* (encoding Cu/Zn superoxide dismutase 1 [SOD1] protein) [9,10,11,12], *TARDP* (encoding for TAR DNA-binding protein 43 [TDP-43]) [13], and *FUS* (encoding fused in sarcoma/translocated in sarcoma [FUS/TLS] protein) have been implicated [14,15]. Moreover, it has been proposed that ALS pathogenesis arises from the collaboration of multiple genes, each acting independently, thereby underscoring a specific genetic susceptibility among ALS patients [5]. External factors also seem to play a pertinent role in triggering neurodegenerative processes linked to ALS, particularly in individuals with a susceptible genetic predisposition [16,17,18]. Indeed, various risk factors have been identified as potential catalysts for ALS onset, encompassing age, smoking, body mass index (BMI), physical fitness level, and exposure to environmental hazards like chemicals, pesticides, metals, and electromagnetic fields, among others [19]. Nevertheless, uncertainties persist regarding the precise contribution of these factors, as the root cause of the disease has not been identified.

To date, a cure for ALS remains elusive and existing therapies primarily address symptom management, yielding minimal impact on survival time. Currently, seven drugs are approved by the US Food and Drugs Administration (FDA): Qalsody (tofersen, # N215887)), RELYVRIO (AMX0035, #N216660), Radicava™ (edaravone; #N209176), Rilutek (riluzole, now generic; #N020599), Tiglutik™ (thickened riluzole; #N209080), Exservan (riluzole oral film; #N212640), and Nuedexta^®^ (#N021879). The recent approval of tofersen, AMX0035, and edaravone have benefited from the application of the American “Accelerating Access to Critical Therapies for ALS Act” (Public Law 117–79). In Europe, to date, only riluzole has been approved by the European Medicines Agency (EMA) (under the reference EMEA/H/C/000109).

ALS is marked by the progressive degeneration of both upper motor neurons (UMN) in the cerebral cortex and lower motor neurons (LMN) in the brain stem and spinal cord. This degeneration precipitates muscle weakness, which gradually evolves into muscle atrophy and paralysis, ultimately culminating in respiratory failure and death [4,20]. Increasing evidence suggests that fALS and sALS may share common pathological mechanisms due to their analogous clinical presentations [2,21]. A multitude of molecular pathways potentially underlie disease development, encompassing disrupted RNA metabolism, glutamate excitotoxicity, protein misfolding and aggregation, endoplasmic reticulum stress, impaired protein trafficking, compromised axonal transport, oxidative stress, inflammation, and mitochondrial dysfunction [2,3,22,23]. Much like numerous other neurodegenerative diseases, as mentioned earlier, mitochondrial dysfunctions assume a significant role in ALS progression. Indeed, various mitochondrial abnormalities have been described not only within the central nervous system (CNS) of ALS patients but also in peripheral tissues, such as skeletal muscle [24,25], liver [26], and lymphocytes [23]. These perturbations encompass mitochondrial bioenergetics dysfunction, inefficient calcium buffering, initiation of mitochondrial-related apoptosis, and aberrations in mitochondrial transport and morphology, among other aspects [27,28]. However, despite the compelling evidence of multifaceted mitochondrial dysfunction in ALS, the precise extent of its involvement, and whether it acts as a triggering factor or is a consequence of other intracellular toxic mechanisms, remains enigmatic.

This review seeks to synthesize recent insights into the role of mitochondria and mitochondria-mediated mechanisms in driving cellular damage. Additionally, it aims to offer a deeper understanding of the interplay between different cell types implicated in ALS. Ultimately, only through a more thorough comprehension of these intricate processes can effective mitochondria-targeted therapies be identified for potential pharmacological interventions in ALS treatment.

## 2. Mitochondrial Dysfunctions in ALS

Mitochondria are highly dynamic organelles, vital for normal metabolism and cellular survival, that participate in a diverse array of cellular functions, including energy metabolism, calcium homeostasis, redox regulation, and cell-death mechanisms [2]. Much like in other neurodegenerative diseases, disruptions to mitochondrial structure, dynamics, bioenergetics, and calcium buffering have been extensively documented in ALS patients, as well as in in vitro and in vivo models of ALS [2,29]. Varied morphological changes have been observed in the early stages of ALS disease within motor neurons (MNs) from transgenic mouse models with *SOD1* [30,31,32], *FUS* [33], and *TDP-43* [34,35] mutations; cortical neurons from the C9orf72 mouse model [36]; and muscle samples from ALS patients [24]. These findings underscore the involvement of mitochondria right from the onset of ALS, manifesting not only in the CNS but also in muscle tissue. The indispensable role of mitochondria in driving ALS progression is further underscored by the identification of specific mitochondrial components interacting with ALS-associated proteins. Notable examples include B-cell lymphoma 2 (Bcl-2) [37], voltage-dependent anion channel 1 (VDAC1) [38], and the mitochondrial form of lysyl-tRNA synthetase [38,39], which interact with mutant SOD1; mRNAs encoding mitochondrial respiratory chain components, which interact with TDP-43 [40] and FUS [41] proteins; and VDAC3, a translocase of the inner mitochondrial membrane 50 (TIMM50) [42] and mitochondrial ribosomal proteins [43], which interact with C9orf72. However, the precise extent of the mitochondrial role in driving disease progression remains an enigma.

### 2.1. Alterations in Mitochondrial Respiration and ATP Production

The most well-known function of mitochondria is their generation of approximately 90% of cellular energy in the form of adenosine triphosphate (ATP), an essential currency for virtually every facet of cellular activity. This energy synthesis stems from the intricate process of oxidative phosphorylation (OXPHOS) taking place in the electron transport chain (ETC), situated within the inner mitochondrial membrane (IMM) [44] (Figure 1).

Discrepant findings have emerged concerning mitochondrial respiration and ATP production across distinct ALS models. Examination of postmortem spinal cord tissues from sALS patients unveiled mitochondrial dysfunction characterized by diminished activity of complexes I, II, III, and IV in tandem with decreased mitochondrial DNA (mtDNA) content and citrate synthase activity [45]. Conversely, no discernable alteration in ETC activity surfaced within crude mitochondrial preparations obtained from the motor and parietal cortices and cerebellum of sALS patients [46]. This dichotomy might signify a more pronounced ETC functional impairment in the spinal cord compared to brain tissues. An alternate interpretation of these contradictory outcomes could be rooted in the limited number of samples analyzed in both studies as well as the inherent heterogeneity found among sALS patients.

Remarkably, beyond the CNS, altered ETC activities have also been documented in sALS patients. Instances include decreased activities of complexes I and IV in skeletal muscle [24,47] and decreased complex I activity coupled with lower ATP levels in lymphocytes [48]. Furthermore, when looking to patient fibroblasts, a noteworthy elevation in mitochondrial membrane potential (ΔΨ_m_) has been observed [49,50]. This potentially alludes to a compensatory mechanism aimed at ameliorating inefficient ATP synthesis. However, comprehensive investigations are imperative to validate these alterations.

Disruption in mitochondrial respiration has been extensively documented in mouse models carrying mutated SOD (mutSOD) [46,51,52,53,54,55] as well as in ALS patients harboring mutSOD1 [46,56]. SOD1, a Cu-Zn metalloprotein, plays a pivotal role in converting superoxide anion radicals (O_2_^−^) into molecular oxygen and hydrogen peroxide (H_2_O_2_); it is primarily localized within the cytosol, but also found in the nucleus, peroxisomes, and mitochondria [11]. Although there is evidence of SOD1 accumulation within mitochondria [51,57,58,59,60,61], the precise function of this enzyme within mitochondria remains unclear.

Most studies suggest that both wild-type and mutSOD1 expressed in transgenic mice, particularly SOD1^G93A^, predominantly accumulate in the intermembrane space (IMS) of spinal cord mitochondria [51,57,58,59,61]. However, some reports have also indicated localization on the outer mitochondrial membrane (OMM) [37,62], association with components of the IMM [51], or accumulation in the mitochondrial matrix [63]. Yet, the accumulation of mutSOD1 within the mitochondrial matrix has not been definitively established.

Regarding the impact of mutSOD1 on mitochondrial respiration, studies have revealed the decreased activities of complexes I+III, II+III, and IV in the spinal cord and its ventral horn at the onset of the disease in SOD1^G93A^ mice [51]. Furthermore, a decrease in the activities of complexes I, II, and IV in the spinal cord preceding disease onset was found in the same mouse model [52]. Consistently, decreased mitochondrial respiration rates have been observed in the spinal cord and brain at the pre-symptomatic stage, persisting throughout the disease in SOD1^G93A^ mice. Notably, a decline in complex IV activity in the brain of these mice was associated with cytochrome c-IMM dissociation [53], suggesting an increase in reactive oxygen species (ROS) production leading to enhanced cardiolipin oxidation and subsequent cytochrome *c* dissociation from the IMM. Decreased complex II and IV activities were also documented in SOD1^G93A^ or SOD1^G37R^ in NSC-34 motor-neuron-like cell lines [54], and significant impairment of the complex I-linked OXPHOS, with a lower ΔΨ_m_, was also identified in SOD1^G93A^ NSC-34 cells [55].

Contrasting findings were reported in patients carrying mutSOD1^A4V^, who exhibited increased complex I activity in the frontal cortex [56] and in the motor and parietal cortices [46]. Moreover, elevated activities of complexes II–III were noted in the motor cortex and cerebellum of patients with mutSOD1^A4V^ as well as in the motor and parietal cortices and cerebellum of a single patient with mutSOD1^Ill3T^, with similar findings in the motor cortex and cerebellum of ALS patients without identified SOD1 mutations to date [46]. In this specific context, the heightened activities of the ETC could signify a compensatory response to oxidative damage to the IMM, possibly resulting from OXPHOS uncoupling. In the forebrain of SOD1^G93A^ mice, an elevation in complex I activity was observed during the pre-symptomatic stage when compared to levels in transgenic wild-type counterparts [46]. Discrepancies in these findings may arise from several factors, including differences in experimental conditions, variations among distinct SOD1 mutations, diverse SOD1^G93A^ mouse strains, discrepancies between animal and human physiology, and the disease stage itself. These disparities highlight the intricate nature of alterations in ETC dynamics, which can emerge as either a causal factor or a consequence of oxidative stress mechanisms. In some instances, these alterations may be counterbalanced by the activation of compensatory mechanisms. It is crucial to consider that the SOD1 transgenic animal model features a high copy number of human mutSOD1, rendering it a non-physiological model that may lack many of the phenotypic alterations observed in ALS patients [64,65]. The prevalence of studies documenting alterations in mitochondrial respiration in SOD1 transgenic animal models in contrast to the absence of impairment in mitochondrial respiration observed in studies involving SOD1^A4V^ or SOD1^G93A^ human MNs [66] suggests that mitochondrial dysfunction observed in the SOD1 transgenic animal model may be overestimated and should be carefully interpreted.

Another gene mutation associated with ALS is *TDP-43*, which has demonstrated various cellular respiratory system alterations in both patients and cellular models [67,68]. TDP-43, a DNA- and RNA-binding protein, plays a critical role in RNA processing [69]. Studies have shown that TDP-43 can lead to decreased activity of complex I and to a reduction in ΔΨ_m_. This was associated with increased expression of mitochondrial uncoupling protein 2 (UCP2) in NSC-34 cell lines transfected with *TDP-43* [67]. In neurons from ALS patients, mutant TDP-43 was found to accumulate within mitochondria [40] and to preferentially bind to mitochondrial mRNAs encoding the ND3 and ND6 subunits of complex I. This binding resulted in the disassembly of complex I and decreased ΔΨ_m_ in HEK293 cells overexpressing mutant TDP-43 variants (G298S, A315T, or A382T) or in fibroblasts from ALS patients carrying G298S or A382T mutations [40]. These findings suggest that TDP-43 toxicity directly impacts mitochondrial bioenergetics. However, fibroblasts from ALS/FTD patients carrying TDP-43^A382T^ did not present alterations in mitochondrial respiration when compared to controls, although a decrease in ΔΨ_m_ was observed in these cell lines [70]. FUS, another multifunctional DNA/RNA-binding protein associated with ALS [71], has also been associated with alterations in mitochondrial respiration [41]. Similar to TDP-43, both wild-type FUS and ALS mutant derivatives were found to bind to mRNAs encoding subunits of mitochondrial respiratory chain complexes (RCC) [41]. This binding disrupted mitochondrial networks and led to impaired mitochondrial respiration, both in HEK293T cells overexpressing wild-type FUS and in those expressing mutant FUS^R521C^. Similar effects were observed in fibroblasts derived from ALS patients [41].

As for C9orf72, the most prevalent mutation in ALS, characterized by the expansion of the GGGGCC (G4C2) hexanucleotide repeat in the first intron of the *C9ORF72* gene [72], observations have pointed to reduced activities of mitochondrial complexes I and V within the frontal cortex of 6-month-old C9orf72-ALS/FTD mice [36]. Similarly, a significant reduction in ΔΨ_m_ has been noted in human induced pluripotent stem cells (iPSC)-derived MNs reprogrammed from the fibroblasts of C9orf72 patients [73]. Conversely, C9orf72 fibroblasts from ALS/FTD patients exhibited increased ΔΨ_m_ along with elevated oxygen-consumption rates, heightened ATP levels, greater complex II activity, and enhanced production of ROS compared to controls [70].

Collectively, these findings underscore the presence of mitochondrial respiration impairment in both the CNS and muscles of ALS patients and animal models. Despite variations in results among different studies, it is apparent that these alterations collectively represent processes of mitochondrial dysfunction that contribute to the progression of ALS.

### 2.2. Role of Oxidative Stress Mechanisms

Mitochondria serve both as a source and a target of ROS [74]. Their most studied function is the production of cellular energy in the form of ATP, a process involving the controlled transport of electrons from substrates to molecular oxygen. As previously mentioned, this intricate electron transfer occurs through a series of enzymatic complexes composing the ETC (Figure 1). When electrons inadvertently leak from mitochondrial ETC complexes directly to molecular oxygen, they generate O_2_^−^ [74], which carries out pivotal roles in cellular signaling [75].

Under normal physiological conditions, cells meticulously maintain a defined redox balance by counteracting ROS with the assistance of the cellular antioxidant defense system [76]. Oxidative stress occurs when this balance is disrupted, either due to excessive ROS production, a compromise in antioxidant defenses, or both [77,78]. In times of cellular stress, elevated ROS levels perturb the redox homeostasis, causing damage to essential biomolecules including proteins, DNA, and lipids. This disruption severely impairs organelle function and eventually leads to cell death [79].

A wealth of evidence indicates that both heightened ROS levels and inadequate antioxidant defenses play substantial roles in ALS [41]. Indeed, post-mortem brain tissue [80,81,82], cerebrospinal fluid (CSF) [83,84,85,86], plasma [87], and urine [88] from ALS patients exhibited increased oxidative damage to proteins, lipids, and DNA. In spinal cord tissues from sALS patients, oxidative damage manifested as elevated levels of protein carbonyls [56,81,82,89], 8-hydroxy-2′-deoxyguanosine (8-OHdG) [81,90], 4-hydroxynonenal (4-HNE) protein conjugates [90,91], nitrotyrosine products [92,93,94], and malondialdehyde-modified proteins [81]. Furthermore, sALS patients’ erythrocytes revealed increased lipid peroxidation concurrent with a decrease in glucose-6-phosphate dehydrogenase activity [95] and in antioxidant defenses, including catalase, glutathione reductase, and glutathione (GSH). These effects appear to intensify with the progression of the disease [95].

Given that SOD1 is a Cu-Zn metalloprotein responsible for converting O_2_^.−^ into molecular oxygen and H_2_O_2_, [11] it is highly plausible that ALS-associated mutations in SOD1 may disrupt the delicate balance of ROS and lead to oxidative damage [96]. Indeed, lymphoblasts from ALS patients with or without identified SOD1 mutations and healthy controls showed distinct redox signatures [97]. Although SOD1 is primarily considered a cytosolic enzyme, it has also been found within the IMS, where it may play a crucial role in safeguarding mitochondria against O_2_^.−^, thereby complementing the function of SOD2, which eliminates O_2_^.−^ released into the mitochondrial matrix [9]. However, the precise mechanism by which SOD1 enters mitochondria has not yet been fully elucidated. In yeast, only an immature form of SOD1, lacking both Cu and Zn and existing in a reduced disulfide state, was found to be able to cross the OMM [98]. It is presumed that the SOD1 apoenzyme gains access to the IMS through the TOM complex [99].

Like other IMS proteins, the import and retention of SOD1 within the IMS are tightly linked to its folding and maturation processes, which include the formation of disulfide bonds. The cysteine residues within SOD1 play a pivotal role in localizing it to the IMS [98,99]. Moreover, it has been proposed that the IMS import of SOD1 involves a copper chaperone (CCS), while its mitochondrial distribution might be regulated by the Mia40/Erv1 disulfide relay system [99,100]. Despite evidence of SOD1 accumulation in mitochondria, the exact mechanisms through which mutSOD1 induces oxidative stress and mitochondrial damage remain to be fully understood, and it is not clear whether these effects are a cause or a consequence of the disease. Considering that only an immature form of SOD1 can traverse the OMM [98], it is possible that mutSOD1 increases the fraction of intramitochondrial SOD1 due to inefficient cytosolic maturation, potentially explaining the heightened mutSOD1 accumulation under pathological conditions. Additionally, the increased mitochondrial accumulation of both wild-type and mutSOD1 in vitro following experimental copper depletion suggests that heightened mitochondrial mutSOD1 accumulation might also be involved in certain pathological mechanisms associated with this mutation [101]. Despite the uncertainties surrounding the deleterious effects of this mutation, it has been proposed that mutSOD1 exerts its pathological impact through a toxic gain of function mechanism rather than by altering SOD1′s enzymatic activity [102].

Various hypotheses have been put forth to elucidate the toxic effects of mutSOD1. One such hypothesis suggests that mutSOD1 induces oxidative damage through an abnormal association with zinc [103] or through exposure to toxic copper at the active site, leading to an elevation in O_2_^.−^ levels [104]. Additionally, it has been proposed that mutSOD1 can exert its toxicity through the following mechanisms: (1) peroxidase activity–mutSOD1 may function as a peroxidase, demonstrating reverse activity, by utilizing H_2_O_2_ as a substrate [58,105]; (2) reactivity with peroxynitrite–mutSOD1 might react with peroxynitrite, resulting in tyrosine nitration [77,106,107]; and (3) formation of soluble aggregates–misfolded SOD1 can form soluble aggregates, leading to decreased stability of SOD1 monomer/dimers [108]. Results observed in MNs derived from iPSCs showed that SOD1 aggregates are progressively and stably formed and transported to neighboring cells [109]. Indeed, there is evidence of increased susceptibility to oxidative stress and mitochondrial damage attributed to misfolded SOD1. This has been observed in mitochondria isolated from the spinal cords of SOD1^G93A^ rats and SOD1^G37R^ mice [110]. Furthermore, in the NSC-34 mouse motoneuronal cell line, mutSOD1 was found to accumulate in an oxidized and aggregated state. This accumulation appears to be associated with mutSOD1 interaction with mitochondria and the oxidation of cysteine residues. Thus, mutSOD1 may consequently contribute to respiratory chain impairment and an imbalance in the ratio of reduced glutathione (GSH) to oxidized glutathione (GSSG), shifting towards a more oxidizing environment [111]. Supporting this, decreased GSH levels and elevated GSSG levels were observed in the lumbar tissues of the spinal cords of mutant SOD^G93A^ mice. Additionally, genetic induction of diminished antioxidant defenses led to notable MN degeneration in hSOD1^WT^ mice, a hemizygous mice model with moderate over-expression of wild-type hSOD1 that does not typically exhibit paralysis or a shortened lifespan [112]. A similar approach was applied to investigate the effects of decreased antioxidant defenses in SOD1^G93A^ mice, which overexpress ALS-linked mutSOD1 [113]. In this case, a 70% decrease in GSH levels further exacerbated the detrimental impact of mutSOD1 on life span. This was associated with increased oxidative stress, aggravated mitochondrial dysfunction, and an elevated association of mutSOD1 with mitochondria [113]. Further evidence of mutSOD1′s negative effects on redox balance has been observed in SH-SY5Y human neuroblastoma cells overexpressing mutSOD1 [114]. Here, it was noted that mutSOD1 induced the activation of Src homology 2 domain containing (Shc) transforming protein 1 (p66Shc). p66Shc is an alternatively-spliced isoform of a growth factor adapter that is phosphorylated upon oxidative stress, resulting in its translocation to mitochondria, where it binds to cytochrome c and, thereby, functions as an oxidoreductase and amplifies ROS production [114].

Apart from mitochondria, another significant source of ROS within the cell stems from the activity of adenine dinucleotide phosphate (NADPH) oxidases (NOX). These NOX enzymes form a family of enzyme complexes that facilitate the transfer of electrons from NADPH to molecular oxygen, ultimately generating O_2_^.−^ and H_2_O_2_ [115]. A pivotal feed-forward loop of ROS production has been identified, driven by the crosstalk between mitochondria and NOX [115]. Essentially, ROS produced by NOX activity can assail mitochondria, while ROS generated within mitochondria can stimulate NOX activity, culminating in amplified ROS production. Members of the NOX family have emerged as potential sources of ROS in ALS. Specifically, the deletion of *Nox2* and, to a lesser extent, *Nox1* has been shown to decelerate disease progression and enhance the survival of SOD1^G93A^ mice, underscoring the role of NOX activity in the oxidative imbalance observed in ALS [116,117]. This is substantiated by evidence revealing elevated NOX activity, particularly NADPH-dependent O_2_^.−^ production, in SOD1^G93A^ mice. This increase appears to result from the physical interaction of mutSOD1 with Rho-like small GTPases, specifically Rac1, which serves as a regulator of NOX activity [118]. Under non-physiological conditions, such as those encountered in an oxidative environment, the dissociation of mutSOD1 from Rac-GTP is impeded, causing the enzyme to remain in an active state and, consequently, intensifying Nox2-derived O_2_^.−^ production [118]. NADPH oxidase inhibition, particularly with apocynin and its derivative diapocynin, was reported to significantly extend the survival of SOD1^G93A^ ALS mice [118]. However, this was challenged by a follow-up study conducted at two different institutions [119], which failed to replicate the remarkable survival extension previously reported [118]. These results cast some doubt on the role of NADPH oxidase in the SOD1 mouse model of ALS. Moreover, the lack of consistent and significant protective effects in various trials raises important questions about the translational potential of NADPH oxidase inhibition as a therapeutic strategy for ALS.

The significance of disrupted redox homeostasis for ALS was further underscored by a study involving MNs from both sALS and fALS patients as well as rodent mutSOD1 ALS models (H46R/G93A rats and G1H/G1L-G93A mice) [120]. As the disease progressed, and concurrent with the decline in spinal MNs, a diminishing number of neurons was found to retain peroxiredoxin-II (PrxII) and glutathione peroxidase-l (GPxl), which are key regulators of the redox system [120]. However, during disease progression, a small population of MNs was also observed to exhibit overexpression of PrxII/GPxl, indicating an up-regulation of the redox system in these cells. This suggests that some neurons retain the ability to respond to redox stress, at least until the advanced stages of the disease [120].

Another vitally important antioxidant pathway that responds to oxidative stress is the nuclear factor erythroid-2-related factor 2 (Nrf2) pathway. In balanced conditions, Nrf2 is sequestered in the cytosol by Keap1 (Kelch ECH-associating protein 1), a cytoplasmic regulator that acts as a sensor for ROS. When ROS levels increase, cysteine residues on Keap1 undergo oxidative modification, releasing Nrf2. This liberated Nrf2 then translocates to the nucleus, where it upregulates the expression of antioxidant genes. These include *GPX*, catalase, and enzymes involved in glutathione metabolism, such as glutathione S-transferase, glutathione cysteine ligase modifier subunit, and glutathione cysteine ligase catalytic subunit (GCLC) [121,122,123,124,125]. Notably, *SOD1* is also among the genes upregulated by Nrf2, making this pathway particularly intriguing for ALS research, especially in cases associated with SOD1 gene mutations.

Despite the precise role of Nrf2 in ALS pathogenesis remaining unclear, accumulating evidence suggests that Nrf2 plays a pivotal role, and its activity may serve as a novel therapeutic target [22]. In both NSC34 cells expressing mutSOD1 and MNs isolated from familial SOD1-associated ALS patients, Nrf2 levels have been shown to be lower than those found in control conditions [126,127]. This implies a disturbance in the Nrf2 pathway and a defective response to oxidative stress. It is important to note that Nrf2 levels may fluctuate throughout the course of the disease. Interestingly, the Nrf2 pathway is naturally activated in muscle tissue during the onset of pathology in the SOD^G93A^ mouse model [128]. Dysregulation of the Nrf2 pathway has also been observed in sALS patients who do not express mutSOD1, suggesting that Nrf2 may play a role in ALS independently of mutSOD1 [128]. Further research is needed to fully comprehend the mechanistic underpinnings of Nrf2 pathway activation and impairment in ALS.

In the context of the redox system, catalase, an enzyme responsible for converting H_2_O_2_ into water and oxygen, which is normally absent from mitochondria, was found to enhance mitochondrial antioxidant defenses and mitochondrial function in primary cultures of astrocytes carrying SOD1^G93A^ when genetically modified to target mitochondria [129]. However, overexpression of catalase in SOD1G93A mice had no discernible effect on their lifespan, indicating that intervention in this pathway alone is insufficient to halt disease progression [129].

Oxidative damage in ALS has also been attributed to other mutated ALS-linked genes. For instance, mutations in TDP-43, a protein belonging to the heterogeneous nuclear ribonucleoproteins (hnRNP) family, have been found to disrupt the Nrf2 pathway [130,131,132]. This disruption likely occurs through the interaction of TDP-43 with another hnRNP family member, hnRNP K, which influences its expression and the RNA-binding properties of the protein [131], subsequently affecting its antioxidant gene transcripts’ binding and function. Consequently, while Nrf2 signaling remains activated in response to stress, its function is compromised due to the abnormal binding of hnRNP K to antioxidant gene transcripts, leading to uncontrolled oxidative damage and toxicity [131]. Multiple lines of evidence support the association between TDP-43 and the dysregulated Nrf2 pathway.

In fibroblasts from patients with the TDP-^43M337V^ mutation and in astrocyte cultures from TDP-43^Q331K^ mice, GSH levels (downstream of Nrf2 activation) were found to be decreased, indicating an impaired oxidative stress response. Moreover, NSC-34 cells overexpressing TDP-43^M337V^ showed a decline in antioxidant defenses, leading to increased intracellular lipid peroxidation, lower cell viability, nuclear accumulation of Nrf2, and decreased protein expression of NAD(P)H quinone dehydrogenase 1 (NQO1, downstream of Nrf2 signaling) [130]. Similar results were obtained in another study using cells expressing TDP43 with two mutations (M337V/Q331K) [132].

Regarding C9orf72, few studies directly implicate oxidative damage as a causative factor in the disease. Nevertheless, evidence suggests a connection between oxidative stress and ALS in the context of C9orf72 mutations. Oxidative stress and DNA damage were found to be increased in iPSC-derived C9orf72 MNs [43]. Additionally, another study demonstrated that dysfunction in iPSC-derived astrocytes from ALS patients with mutated C9orf72 was also associated with elevated oxidative stress. Interestingly, the medium from these cell cultures induced oxidative stress in wild-type MNs [133].

While substantial evidence strongly underscores the contribution of excessive ROS in ALS, which are primarily generated within mitochondria, as well as compromised antioxidant defenses [22,134], the precise role of oxidative damage in the development and progression of the disease remains a subject of inquiry. Specifically, it remains to be elucidated whether oxidative damage serves as a primary instigator of the disease or arises as a secondary consequence stemming from the initial toxic mechanisms that trigger the onset of ALS.

### 2.3. Metabolic Dysregulation

The investigation of energy homeostasis in ALS research has revealed significant dysregulation in both ALS patients [135] and in models of the disease involving SOD1 and TDP-43 [136,137,138,139,140]. ALS patients exhibited an imbalance between food intake and energy expenditure [141], and research has shown increased energy demands in the muscles of SOD1 (G86R and G93A) mice compared to controls [136]. Interestingly, by using mice expressing human SOD1 exclusively in skeletal muscle, Martin and Wong showed that the presence of hSOD1 only in skeletal muscle resulted not only in limb weakness and muscle wasting but also in abnormalities in adipose tissue, suggesting a relationship between muscle and adipose tissue in the context of ALS disease [142].

Malnutrition is a recognized problem among ALS patients, with its prevalence ranging from 15% to 55%, and it is negatively correlated with disease severity [143,144,145,146]. There is substantial evidence suggesting that malnutrition serves as a prognostic factor for poorer survival rates among ALS patients [144,145,146,147,148]. The progression of malnutrition typically leads to weight loss [147,149], attributable to dysphagia resulting from bulbar weakness or heightened metabolic activity [148,150,151,152].

Interestingly, studies have indicated that individuals with a lower BMI are at an increased risk of developing ALS [153], while, conversely, overweight individuals face a lower risk of up to 40% [154]. Consequently, weight loss emerges as a crucial prognostic factor for patients [155] and can serve as a valuable clinical indicator of the necessity for nutritional support.

Several studies have reported instances of hypermetabolism, characterized by an increase in measured metabolic rate compared to calculated values, in a significant proportion of ALS patients [144,148,149,150,151,156,157,158,159]. Many of these studies have applied equations for predicting resting energy expenditure [160], although these equations still require validation for ALS [161] and often do not account for the muscle atrophy frequently observed in ALS patients [159]. The prevalence of hypermetabolism in sALS patients has been documented to range from 25% to 68% [144,148,149,150,151,158,159], while it was reportedly found in 100% of fALS individuals [156]. Notably, approximately 80% of ALS patients in one study were found to maintain a consistent metabolic phenotype over time, and this phenotype showed no correlation with disease characteristics [148]. However, it was observed that hypermetabolism negatively correlated with age but positively correlated with BMI and fat-free mass [148]. In contrast, some studies have reported an increase in hypermetabolism as patients neared the end of life [149], although this finding was not replicated in other research [148,150,151]. Nevertheless, a trend towards worse survival among hypermetabolic ALS patients was subsequently noted [158]. Further insights come from reports of increased prevalence of hypermetabolism in ALS patients compared to age- and sex-matched control individuals [159]. In this study, hypermetabolism was associated with a significant reduction in fat-free mass in ALS patients [159]. Notably, this study was the first to consider the muscle atrophy observed in these patients, reporting a prevalence of 41% hypermetabolism in ALS patients compared to 12% in healthy, matched subjects [159]. Importantly, hypermetabolic ALS individuals in this study experienced a significantly faster rate of functional decline and worse survival than normometabolic ALS patients [159], underscoring the potential utility of metabolic indices in predicting ALS outcomes. However, a systematic review [162] analyzing 29 selected articles from various databases and aiming to correlate changes in metabolic parameters with ALS progression and survival concluded that the use of different metabolic parameters and methodologies across studies made it challenging to establish correlations [162]. The parameters examined varied widely, including BMI (65.5%), hypertension (20.7%), cardiovascular risk (6.89%), obesity (10.3%), diabetes (13.8%), and glycaemia (10.3%) [162]. Despite the assumption that metabolism influences ALS, the study could not definitively establish an association between metabolism and disease development [162]. The authors emphasized the need for standardized methodologies and the analysis of metabolic parameters across different studies to yield more conclusive results [162].

The dysregulation of metabolic pathways associated with stress and oxidative responses may contribute significantly to the progression of ALS [163]. Despite extensive research, the root causes of metabolic dysregulation and its effects on neuronal metabolism in ALS patients remain incompletely understood. Nevertheless, mitochondria have emerged as central players in this process [148]. Numerous studies employing proteomic approaches have reported substantial alterations in proteins linked to glycolysis, mitochondrial function, and stress responses, particularly in SOD1^G93A^ mice when compared to control animals [163,164,165,166,167,168]. A recent study employed targeted proteomics to quantify proteins involved in metabolic pathways at the pre-onset, onset, and end stages of the disease in the spinal cord of SOD1^G93A^ mice [163]. Their findings revealed significant changes in metabolic protein profiles, even at early stages preceding disease onset. Notably, an increased abundance of phosphofructokinase and lactate dehydrogenase—crucial enzymes regulating glycolysis and tricarboxylic acid (TCA) cycle rates, respectively—was observed [163]. These results align with a prior study that reported heightened glycolytic flux in a neuronal cell line expressing SOD1^G93A^ [169]. Moreover, increased glucose uptake was detected in the spinal cords of transgenic SOD1^G93A^ mice during the pre-symptomatic stage, which progressively declined as the disease advanced [170]. Importantly, an elevated rate of glycolysis can impact other metabolic pathways, including the pentose phosphate pathway, potentially decreasing cellular antioxidant capacity and triggering cellular apoptosis [171]. At the later stages of the disease, reductions in the levels of various glycolytic proteins (aldolase, glyceraldehyde phosphatase, and phosphoglycerate kinase), which are important proteins involved in the malate–aspartate shuttle (cytoplasmic and mitochondrial aspartate aminotransferases, and mitochondrial malate dehydrogenase), pyruvate dehydrogenase, and TCA cycle enzymes (malate and isocitrate dehydrogenase-related proteins) were reported [163]. Conversely, certain proteins related to fatty acid metabolism and the ETC exhibited increased abundance in the spinal cords of SOD1^G93A^ mice [163]. These alterations in protein content are consistent with a shift from glycolysis to oxidative metabolism, which was also been reported in the muscles of SOD1 mice [172,173], which have higher energetic needs [136]. This shift in metabolic strategy is probably driven by the heightened energy demands of large MNs, which are particularly susceptible to changes in metabolism [163]. Interestingly, the glycolytic ATP production rate showed potential diagnostic and therapeutic interest in lymphoblasts from ALS patients [174].

When glucose demand outstrips the available supply, neurons begin to utilize ketone bodies produced by the liver and, notably, by astrocytes [175,176,177,178,179]. The production of ketone bodies by astrocytes, which enhances oxidative metabolism, is reflected in the upregulation of proteins related to β-oxidation, fatty acid metabolism [137,163], and the levels of 3-hydroxy-3-methylglutaryl-CoA lyase, a key enzyme in the ketogenic process [179]. Dysregulated fatty acid metabolism has been reported in both ALS animal models and ALS patients, with evidence suggesting that dyslipidemias play a pivotal role in disease progression [141].

Positive protein inclusions of TDP-43 are an almost ubiquitous feature in ALS cases, establishing them as a fundamental pathological hallmark of the disease [180,181]. Through metabolic profiling, significant changes in glycolysis and pentose phosphate pathways were found in Drosophila models of TDP-43 proteinopathy [140]. The TDP-43 mutants exhibited increased levels of pyruvate, suggesting an upsurge in glycolytic activity and a substantial upregulation of phosphofructokinase, the enzyme governing the rate-limiting step in glycolysis [140]. These findings align with previous research showing elevated pyruvate levels in the plasma of ALS patients [182,183]. The Drosophila TDP-43 mutants also presented an intensified glycolytic input into the pentose phosphate pathway and heightened levels of glucose-6-phosphate dehydrogenase, potentially providing a mechanism to counteract oxidative stress by enhancing NADPH production [140].

Another study adopted a metabolic profiling approach using astrocytes reprogrammed from fibroblasts obtained from C9orf72, sALS, and age-matched control subjects to explore C9orf72-dependent metabolic dysfunctions [184]. This study unveiled distinct metabolic profiles and the loss of metabolic flexibility in C9orf72 and sALS cells in comparison to controls [184]. Specifically, deficiencies were noted in the metabolism of adenosine, fructose, and glycogen along with disruptions in the membrane transport of mitochondrial substrates, contributing to the loss of metabolic flexibility in C9orf72 and sALS cells. These findings underscore the importance of metabolic flexibility in scenarios characterized by bioenergetic stress [184].

Metabolomic investigations have identified distinct metabolite profiles in samples from both ALS patients and mouse models when compared to controls [182,183,185,186,187,188,189,190,191,192]. These studies have revealed alterations in metabolites linked to glucose metabolism [185,189,190,191], with correlations established between these changes and ALS disease progression [186,190,191]. Some preliminary studies have even explored the potential of these metabolic profiles to predict ALS onset [182]. Additionally, elevated levels of ketone bodies have been observed in the cerebrospinal fluid of ALS patients, suggesting potential alterations in lipid beta oxidation [185,186]. Notably, distinct metabolite profiles have been observed among different subtypes of ALS patients, underscoring the heterogeneity of the disease [187,188].

### 2.4. Mitochondrial Dynamics and Biogenesis

Mitochondrial dynamics and biogenesis are tightly regulated by a complex network of signaling pathways within cells [193]. Mitochondria exist as part of a highly dynamic network that relies on a delicate balance between fission and fusion events to maintain their functional integrity. These processes control mitochondrial size, number, and shape, facilitating the mixing of mitochondrial contents, including proteins, lipids, and DNA [194,195]. Fission plays a crucial role in eliminating dysfunctional mitochondria, while fusion enables the intermingling of contents from both functional and defective mitochondria, thereby preserving overall mitochondrial integrity [196].

The key regulator of mitochondrial fission is dynamin-related protein 1 (Drp1), which becomes phosphorylated and relocates from the cytosol to the OMM when a mitochondrion is targeted for fission [197,198]. Drp1 is recruited by various receptor proteins, including mitochondrial fission factor (Mff) [199], mitochondrial fission 1 protein (Fis1) [200], and mitochondrial dynamics proteins of 49 and 51 kDa (MiD49/51) [201]. Following recruitment, Drp1 oligomerizes and forms a ring-like structure through self-assembly and GTP hydrolysis, and this structure severs the mitochondrial membrane, leading to mitochondrial fragmentation [202,203].

Conversely, the process of mitochondrial fusion involves both the IMM and the OMM. Major players in OMM fusing are mitofusins 1 and 2 (Mfn1 and Mfn2), while optic atrophy protein 1 (OPA1) governs the IMM fusion [204,205]. Mfn1 and Mfn2, situated in the OMM, facilitate the fusion of mitochondrial membranes. OPA1, located in the IMM, interacts with both Mfn isoforms, forming intermembrane protein complexes that orchestrate the fusion of the OMM to the IMM [206,207]. Given the critical dependency and sensitivity of neurons to alterations in mitochondrial dynamics, several studies have underscored the link between the disruption of mitochondrial dynamics regulators, such as Drp1, Mfn2, and OPA1, and the emergence of dysfunctional mitochondria, often leading to neurodegenerative processes [208,209,210,211].

The disruption of mitochondrial dynamics has been extensively documented in both animal models [31,34,35,212,213] and ALS patients [214]; however, the precise mechanisms underlying these alterations are still a subject of debate. While specific changes may vary among different ALS models, most studies consistently reveal an increase in mitochondrial fission relative to fusion [30,31,215]. In the case of SOD1^G93A^ mice, researchers noted an initial increase in the mitochondrial fusion proteins Mfn1 and OPA1 during the early stages of the disease, followed by a significant decrease in later stages [31]. This biphasic pattern could suggest an initial protective response involving mitochondrial fusion followed by a gradual shift towards increased fission. Lines of evidence have revealed that mitochondrial dynamics are also under cellular redox control. It was also already discussed here that mutSOD1 may contribute to an imbalance in the GSH:GSSG ratio towards a more oxidizing environment [111]. Using HeLa cells, Shutt and collaborators have shown that GSSG strongly induces mitochondrial fusion by inducing the formation of disulfide-mediated Mfn oligomers in a process requiring GTP hydrolysis, whereas neutralizing oxidative stress with GSH or ROS scavengers impaired the fusion process [216]. On the other hand, a study conducted using endothelial cells showed that the depletion of the mitochondrial-located protein disulfide isomerase A1, an oxidoreductase with thiol reductase activity for Drp1, induced a ROS-mediated protein post-translational modification in which a reactive cysteine thiol is transformed into cysteine sulfenic acid. This Drp1 post-translational modification, termed protein sulfenyltion, is a key initial readout of redox signaling that ultimately leads to mitochondrial fragmentation, further ROS increases, and mitochondrial respiratory impairment [217]. This way, one can hypothesize that this or similar ROS-mediated sulfenylation processes can occur in ALS and further increase mitochondrial fission and its detrimental effects. Additionally, mitochondrial fission-related proteins, such as pSer637-Drp1 and Fis1, were found to increase during the early stages of the disease, potentially targeting cells for apoptosis and mitophagy [31]. Similar alterations in mitochondrial dynamics were observed in skeletal muscle biopsies from ALS patients [214] and in the MNs of rat spinal cords [212].

Although less extensively studied, models of ALS associated with TDP-43 [34,35,213], FUS [218,219], and C9orf72 [70] mutations have also exhibited evidence of mitochondrial fragmentation. For instance, in TDP-43 models, there was an increase in Fis1 protein levels and phosphorylated Drp1 along with decreased Mfn1 protein expression in the brain lysates of transgenic (TDP^43PrP^) mice expressing full-length human TDP-43 (hTDP-43) driven by the mouse prion promoter [213]. Moreover, OPA1 protein expression was found to be increased in brain lysates of 3-month-old TDP-43^A315TKi^ animals, decreasing with age, particularly at 15 months of age. This suggests that young TDP-43^A315TKi^ ALS mice experience enhanced mitochondrial fusion that shifts toward greater mitochondrial fragmentation as they age [34]. Interestingly, overexpression of mutant TDP-43 (Q331K and M337V) in cultured neurons led to a shift in the mitochondrial fission–fusion balance towards fission, causing mitochondrial fragmentation in the dendrites and cell bodies of MNs rather than in axons. Conversely, TDP-43 knockdown shifted the balance towards fusion, resulting in mitochondrial elongation. These findings indicate a specific role of TDP-43 in regulating mitochondrial dynamics [35]. Mitochondrial fragmentation has also been reported in MNs expressing either R521G or R521H FUS [218] and in HT-22 mouse neuronal-like cells expressing the P525L-mutant FUS [219] compared to neurons expressing wild-type FUS. Similarly, FUS transgenic flies exhibited mitochondrial fragmentation compared to wild-type flies [219]. Additionally, a reduction in the length of mitochondria was observed in fibroblasts from patients expressing ALS mutant C9orf72, suggesting a disruption in mitochondrial dynamic balance driven by the most prevalent ALS mutation [70].

Mitochondrial biogenesis plays a pivotal role in maintaining mitochondrial homeostasis throughout the cellular life cycle, ensuring that the physiological demands of eukaryotic cells are met by synthesizing new mitochondrial components to replace damaged mitochondria [29,220]. This tightly regulated process involves the coordinated efforts of both mtDNA and the nuclear genome to create new mitochondria [221]. One of the principal regulators of mitochondrial biogenesis is peroxisome proliferator-activated receptor gamma coactivator 1-alpha (PGC-1α), which stimulates mitochondrial biogenesis by activating various transcription factors, including nuclear respiratory factor 1 (NRF1) and nuclear respiratory factor 2 (NRF2, also known as GA-binding protein-GABP and not to be confused with Nrf-2) [29,220]. These transcription factors, in turn, activate mitochondrial transcription factor A (Tfam), which is responsible for the transcription of nuclear-encoded mitochondrial proteins and the maintenance and replication of mtDNA [29].

Numerous studies have provided evidence of altered mitochondrial biogenesis markers in both ALS mouse models and patients [222,223,224]. In the mouse model carrying SOD1^G93A^, research has revealed diminished levels of PGC-1α mRNA in the spinal cord [222,223,224] and skeletal muscle [214,224]. Notably, increased levels of PGC-1α mRNA were reported in the skeletal muscle of the same mouse model [222]. Furthermore, protein expression levels of PGC-1α were found to be reduced in the skeletal muscle of the SOD1^G93A^ model [214,224] and in the skeletal muscle of ALS patients [214]. Collectively, these findings suggest that both PGC-1α mRNA and protein expression may play a role in the pathogenesis of the disease, particularly in the context of SOD1 mutations. In this same model, Tfam mRNA levels were shown to decrease in the spinal cord and skeletal muscle [224]. Additionally, protein levels of NRF1 were found to be diminished in the skeletal muscle of the SOD1^G93A^ mouse model [214,224] and in the skeletal muscle of ALS patients [214]. Taken together, these observations of varying protein and mRNA levels of mitochondrial markers in both animal models and patients, particularly in the spinal cord and skeletal muscle of individuals with SOD1 mutations, indicate a potential disruption in the mitochondrial biogenesis process affecting the cellular capacity to maintain a healthy network of functional mitochondria, which further underscoring mitochondrial relevance in ALS pathogenesis. Interestingly, Tfam protein levels in the lymphoblasts of ALS patients are of potential diagnostic and therapeutic interest [174].

Several studies have reported variations in mtDNA levels in both ALS mouse models and patients, shedding light on the complexity of mitochondrial dynamics [225,226]. For instance, the SOD1 knockout mouse model (SOD^−/−^) exhibited heightened mtDNA levels in skeletal muscle [225]. In patients—notably those with SOD1 and C9orf72 mutations—peripheral blood mtDNA levels were also found to be elevated [226]. Additionally, fibroblasts from C9orf72 patients displayed increased mtDNA levels and mitochondrial mass [70]. These observed increases in mtDNA content likely represent a compensatory mechanism in response to the abnormal mitochondrial function observed in these individuals [70,226]. Conversely, fibroblasts from patients harboring TARBDP mutations exhibited reduced mitochondrial mass, without alteration in mtDNA levels [70]. This suggests that TARBDP mutant fibroblasts might not possess the capacity to activate mitochondrial proliferation mechanisms to counterbalance dysfunctional mitochondria.

In summary, various studies indicate that mtDNA levels are subject to alterations, spanning from mouse models to human patients, hinting at their potential as a marker for the pathological processes underlying ALS. Furthermore, these mtDNA alterations seem to differ when comparing different mutations [70,226], especially between mutSOD1 and C9orf72 patients versus TARBDP patients. As a result, these findings prompt speculation that distinct mutations may induce specific modifications in mitochondrial biogenesis markers, although further research is needed to elucidate the precise mechanisms involved.

### 2.5. Mitochondrial Trafficking in ALS

Mitochondrial trafficking in neurons is a highly orchestrated process that varies depending on the direction of the movement, the polarity, and the arrangement of neuronal microtubules (MTs) [227]. MTs are polymeric structures composed of α- and β- tubulin heterodimers with a polar head-to-tail structure. Within MTs, α-tubulin monomers are oriented toward the slower-growing end (the minus end), while β-tubulin faces the faster-growing end (the plus end). In neurons, the minus end faces the cell body and the plus end extends toward the cellular extremities [228,229]. Mitochondria can move in both directions along these MTs. Retrograde transport, from the extremities to the cell body, is powered by dynein motors, whereas the anterograde transport, from the cell body to the extremities, is orchestrated by the kinesin superfamily [230,231]. These movements are energy-dependent, relying on ATP hydrolysis [232].

In addition to the transporting complexes, various protein adaptors play critical roles in the trafficking of neuronal mitochondria. These adaptors include the mitochondrial rho (MIRO)/Milton complex in Drosophila [233,234], trafficking kinesin protein (TRAK) 1 and 2 in mammals [235,236], syntabulin [237], and RAN-binding protein 2 (RANBP2) [238], among others. Beyond MTs, actin filaments and their associated protein motor adaptors also contribute significantly to mitochondrial movement. Myosins 2 and 19, for instance, play crucial roles in the short-distance movement of mitochondria along these filaments [239,240]. During mitochondrial transit, there are instances when they must halt to facilitate proper axonal and synaptic functions, which necessitates dissociation from the motor proteins and docking [241]. Syntaphilin appears to be a key protein involved in this process, immobilizing mitochondria by anchoring them to the MTs (Figure 2) [242,243].

While the precise biological and molecular origins of the phenomenon remain a topic of significant debate, the defective axonal trafficking of mitochondria stands out as one of the earliest neuropathological features observed in ALS models [244]. Various lines of evidence have consistently demonstrated that alterations in mitochondrial trafficking and morphology occur within the sciatic nerve of ALS SOD1 mutant mice. Notably, these abnormalities manifest in the early stages of ALS, even before any clinical symptoms become apparent. This malfunction typically commences with disruptions in retrograde mitochondrial transport, which precede defects in anterograde transport and ultimately culminate in mitochondrial fragmentation, as also observed in TDP43 ALS mutant mice [245]. These retrograde transport impairments have been associated with the inadequate removal of damaged mitochondria via mitophagy [246,247]. In both ALS models, anomalies in anterograde transport appeared at later stages of the disease and were linked to abnormal mitochondrial clustering (Figure 2) [245]. The net increase in the retrograde trafficking of mitochondria may signify an enhanced clearance of damaged mitochondria via mitophagy [248]. Nevertheless, it is important to note that in rat cortical neurons transfected with mutSOD1, a different scenario unfolds; these neurons exhibited decreased anterograde transport of mitochondria, without concurrent retrograde transport issues. This disturbance resulted in increased mitochondrial content in cell bodies compared to the cell axons, potentially triggering axonal stress by limiting mitochondrial Ca^2+^ buffering and diminishing ATP synthesis derived from mitochondria (Figure 2) [249,250].

Exploring the molecular mechanisms underlying the reduction of anterograde transport in cortical neurons from the E18 embryos of ALS SOD1 mutant rats, researchers have identified a decreased expression of Miro protein in these neurons. This is a hallmark of PINK1/Parkin-dependent mitophagy [248], where PINK1 phosphorylates Miro, targeting it for Parkin-dependent degradation [251] and ultimately leading to the cessation of mitochondrial transport and the buildup of damaged mitochondria. Interestingly, it has been found that either the restoration of Miro1 expression or the knockdown of Pink1 can restore mitochondrial motility [248]. Additionally, various causes of alterations in mitochondrial have been proposed, including pathogenic mutations affecting the axonal transport machinery itself, such as kinesin-1, dynein, Profilin 1, Dynactin, and α-tubulin [244,252,253,254]. Furthermore, indirect processes appear to contribute significantly to this phenomenon, involving pathogenic kinase signaling, microtubule destabilization, and protein aggregation [244].

It is essential to emphasize that different mutant models yield distinct effects, underscoring the significance of recognizing that diverse alterations may arise when utilizing different biological models and cell types.

### 2.6. Mitochondria and Endoplasmic Reticulum Crosstalk

Mitochondria play a significant role in managing Ca^2+^ levels, influencing energy metabolism and cellular activities. This is especially relevant for neurons given that mitochondria localize in neuronal cell bodies and in dendrites, axons, and synaptic terminals. When there is a high concentration of Ca^2+^ in microdomains near mitochondria, the organelle efficiently accumulates Ca^2+^, primarily through the Ca^2+^ uniporter (MCU). This electrogenic carrier is situated at the IMM and operates based on the negative transmembrane potential of mitochondria. This accumulation is facilitated by microdomains adjacent to endoplasmic reticulum (ER) subcompartments that physically interact with mitochondria, referred to as endoplasmic reticulum–mitochondria-associated membranes (MAMs) (reviewed in [255]). MAMs have also been demonstrated to play a crucial role in regulating autophagy and mitochondrial dynamics through the modulation of Drp1 or Mfn2 [256,257]. These complex structures are composed of numerous proteins from both mitochondria and the ER, with additional components yet to be identified [258]. Intriguingly, mass spectrometric analysis has identified more than 1000 proteins in MAM fragments [259]. As MAMs function as a signaling hub, their effectiveness heavily relies on proteins that perform diverse tasks. These proteins are categorized based on their primary functions. For instance, in the domain of Ca^2+^ transport, inositol 1,4,5-triphosphate receptor (IP3R), VDAC1, 2b isoform of Sarco/Endoplasmic Reticulum Calcium ATPase (SERCA), Sigma-1 receptor (Sigmar1), and glucose-regulated protein 75 (Grp75) [260,261,262,263] have been extensively studied. Regarding autophagy, autophagy-related 14 (ATG14) and autophagy-related 5 (ATG5) play pivotal roles [264]. Tethering proteins responsible for the physical contact between mitochondria and the ER, such as ER protein vesicle-associated membrane protein-associated protein B (VAPB) and protein tyrosine phosphatase interacting protein 51 (PTPIP51) [265], are also noteworthy.

Numerous studies indicate that ALS-associated mutations are linked to impaired enzymatic activities within MAM domains, specifically impacting calcium transfer between the ER and mitochondria (reviewed in [266]). Notably, proteins like TDP-43 and FUS induce the dissociation of VAPB and PTPIP51 [265,267], thereby modifying calcium transfer dynamics between these organelles [265]. Moreover, mutations in the *VAPB* gene have been demonstrated to induce a familial variant of ALS. This could be attributed to a decrease in the effectiveness of Ca^2+^ elimination, potentially resulting from inadequate energy to power the Ca^2+^ ATPases, leading to the accumulation of surplus Ca^2+^ within mutated neurons and triggering changes in synaptic vesicle release and the development of synapses [268]. These findings are further supported by a study showing that disruption of VAPB–PTPIP51 tethering and the related delivery of Ca^2+^ from ER stores to mitochondria occurs in neurons derived from iPS cells obtained from patients carrying pathogenic c9orf72 expansions associated with ALS/FTD [269]. Recently, Pilotto and colleagues discovered that ER stress in human C9ORF72-ALS/FTD iPSCs and C9orf72 mouse neurons initiates a compensatory mechanism [270]. This involves the upregulation of GRP75 expression at MAMs, effectively counteracting early mitochondrial Ca^2+^ imbalance. Over time, PolyGA inclusions sequester GRP75, resulting in its diminished presence at the MAMs and leading to functional loss and subsequent mitochondrial dysfunction [270].

Several molecules have been investigated as potential therapeutic agents addressing dysfunction at MAMs in ALS (reviewed in [271]). For instance, salubrinal, an ER stress inhibitor, displayed promising results in SOD1^G93A^ mice, alleviating disease progression by enhancing survival rates and promoting neuromuscular junction reinnervation [272]. Additionally, PRE-084, a sigma-1 receptor agonist, demonstrated notable effects, reducing SOD1^G93A^-induced cell death and extending the survival time of mice [273]. Another example is Sephin1, an ER stress inhibitor, which exhibited significant efficacy in preventing motor neuron degeneration in both in vitro and in vivo ALS models [274,275]. These studies highlight diverse strategies and potential therapeutic avenues for addressing MAM dysfunction in ALS.

### 2.7. Autophagy and Mitophagy

Autophagy is a pivotal catabolic process in cellular recycling and the maintenance of intracellular homeostasis that involves the breakdown of unnecessary or damaged cellular components through lysosomal degradation [276]. More specifically, a form of autophagy known as macroautophagy encompasses a series of intricate signaling events that initiate with the sequestration of proteins and damaged organelles into autophagosomes, followed by their digestion via lysosomal hydrolysis [277]. Dysregulation of autophagic flux has been closely associated with various pathological processes [278,279], including neurodegenerative diseases, such as ALS [280].

In ALS, autophagy is known to occur in the MNs of the spinal cord in both postmortem human ALS patients [281,282] and animal models of the disease [283,284,285,286]; however, its precise functional role in disease progression remains elusive. Mutations in genes encoding proteins for crucial selective autophagy receptors, such as optineurin (*OPTN*) [217], p62/Sequestosome-1, and Tank-binding protein (*TBK1*) [287], have been implicated in ALS, contributing to the impairment of autophagic flux. For instance, ALS patients with C9orf72 expansion exhibited p62 inclusions [281,282], and pathogenic mutations in SOD1 were found to lead to polyubiquitinated aggregates of misfolded mutSOD1 that interact with p62, thereby enhancing the association between mutSOD1 and microtubule-associated light chain 3-II (LC3-II), a primary autophagic marker embedded into autophagosomes [288,289].

Moreover, mutations in the *OPTN* gene have been found in ALS patients [217] and colocalize with SOD1 inclusions [290]. Overexpression of missense mutated OPTN resulted in increased LC3-II levels, potentially impairing autophagosome transport or fusion events [291]. ALS cases have been associated with genes involved in vesicular trafficking, including *VCP*, *CHMP2B*, *VAPB*, *ALS2*, and *DCTN1* (genes encoding valosin-containing protein, charged multivesicular body protein 2B, VAMP associated protein B, alsin Rho guanine nucleotide exchange factor, and dynactin subunit 1 protein, respectively), which may influence autophagy flux either directly or indirectly [292].

Several studies underscore the critical role of autophagy in ALS pathogenesis. mutSOD1 mouse models have exhibited hyperactivated autophagy, characterized by AMPK activation and mTOR downregulation [293], along with increased conversion of LC3-I into LC3-II [294], indicative of heightened autophagic vacuole formation [295,296]. These models have also shown elevated levels of autophagic regulators, such as TFEB, involved in autophagy initiation and lysosomal biogenesis [297], and Beclin1, an autophagic initiator [298], resulting in increased LC3-II levels [299]. Additionally, mutSOD1′s toxic effect extends to excessive interaction with dynein, a protein responsible for axonal retrograde transport. This overload disrupts autophagosome retrograde transport, contributing to the failure of late-stage fusion steps in autophagy [300].

In the context of TDP-43 and FUS, both proteins, much like SOD1, tend to undergo misfolding and aggregation, a process that coincides with autophagic impairment [301,302]. Notably, previous studies have demonstrated that the pharmacological enhancement of autophagy using rapamycin decreased cell death in neurons carrying FUS mutations [303]. Additionally, it promoted the clearance of TDP-43 aggregates in Drosophila and mouse models, resulting in a reduction of ALS symptoms [304,305].

Concerning C9orf72, human neurons derived from iPSC obtained from individuals with C9orf72 repeat expansions exhibited elevated p62 levels and heightened sensitivity to autophagy inhibitors like chloroquine and 3-methyladenine compared to control neurons [306]. These findings suggest a compromised autophagy function in these neurons. In addition, the loss-of-function of C9orf72 can disrupt the mTOR signaling pathway, inhibiting its activity [307,308]. Furthermore, there is a correlation between C9orf72 loss and an increase in lysosomal biogenesis along with an upsurge in swollen lysosomes [308]. Supporting these observations, C9orf72 KO mice displayed elevated levels of lysosomal markers such as LC3I, LAMP1, and PSAP [309], indicating that C9orf72 likely plays a regulatory role in the autophagy/lysosome pathway.

Although extensive research has explored the autophagy process in ALS, the selective degradation of mitochondria through autophagy, known as mitophagy, has only recently gained attention. Mitophagy is of great significance as it serves to eliminate defective and irreparable mitochondria, preventing their accumulation and potential harm to the cell [310]. Consequently, mitophagy plays a central role in mitochondrial quality control.

In cases of mitochondrial dysfunction, the protein PINK1, which typically relocates from the OMM to the IMM for cleavage under normal circumstances, accumulates on the OMM. This accumulation acts as a signal for the recruitment of the E3 ubiquitin ligase Parkin [311]. PINK1 phosphorylates Parkin and ubiquitin, activating Parkin’s E3 ligase activity [311]. Subsequent pathways have been proposed to describe the connection to the autophagy process, although some uncertainties persist. For instance, parkin-independent mitophagy pathways have been identified, notably through the recruitment of the autophagy receptor OPTN by PINK1 [312]. Nevertheless, during mitophagy, defective mitochondria must either be enveloped by an autophagosome or fused with a lysosome [311]. Of note, mutations associated with ALS in genes such as OPTN and TBK1 have been shown to disrupt mitophagy. This disruption may result in the accumulation of dysfunctional mitochondria, potentially contributing to the neuronal degeneration observed in ALS [313,314].

Recently, a novel concept has emerged regarding the removal of dysfunctional mitochondria within cells, termed “mitoautophagy” [315]. In a groundbreaking study, it was discovered that a substantial number of flawed mitochondria in corticospinal motor neurons (CMNs), particularly in young prpTDP-43^A315T^ and PFN1^G118V^ mice, underwent a self-destructive process without relying on lysosomes or autophagosomes as typically observed in mitophagy [315]. Interestingly, distinct mechanisms of mitochondrial removal were identified in CMNs from SOD1^G93A^ and prpTDP-43^A315T^ mice, with the former being autophagosomal dependent. This suggests that mitoautophagy might be specific to TDP-43-associated pathology [315]. This difference was attributed to the absence of TDP-43 pathology in SOD1^G93A^ mice [315]. As ALS patients with SOD1 mutations also lack TDP-43 pathology [315,316], this raised the possibility that various pathways for mitochondrial removal are activated in MNs, each potentially playing a distinct role in ALS development. Consequently, further research is warranted to comprehensively unravel the intricate relationship between mitochondria and ALS pathological processes.

### 2.8. Apoptotic Mechanisms

The apoptosis in MNs plays a pivotal role in the neurodegeneration observed in ALS. Various apoptotic markers have been identified in postmortem spinal cord tissues of ALS patients [317,318] and ALS mouse models [319]. The activation of several caspases has been closely associated with ALS neurodegeneration, encompassing upstream caspases 1 and 9 as well as downstream executioner caspases 3 and 7 [320,321,322]. Additionally, the release of cytochrome c from mitochondria to the cytosol has been observed in the spinal cords of SOD1^G93A^ transgenic mice, underscoring the clear involvement of the mitochondrial apoptotic pathway in mutSOD1-mediated toxicity [322].

In concert with apoptotic pathways, ROS and calcium overload have emerged as common pathological features in the neurodegenerative process of ALS, implicating mitochondrial dysfunctions and cell-death mechanisms. In terms of calcium uptake, it was noted that mutSOD1 impaired mitochondrial calcium-loading capacity in the brain and spinal cord of G93A mutant transgenic mice early in the disease course, long before motor weakness sets in, and massive neuronal death occurred [323].

With respect to calcium dysregulation, several studies have explored the role of the mitochondrial permeability transition pore (mPTP) in the degeneration of MNs in ALS. The mPTP is a megachannel located in the IMM known to function in both physiological and pathological conditions [324], including various neurodegenerative disorders [325]. While under normal physiological conditions, the mPTP serves as a rapid mitochondrial efflux mechanism for managing ROS and calcium overload, under pathological conditions, prolonged mPTP opening can lead to rapid mitochondrial swelling driven by the osmotic pressure of matrix solutes. This ultimately results in the rupture of the OMM and the collapse of ΔΨ_m_, culminating in apoptotic cell death when sufficient ATP is available or necrosis when ATP levels are insufficient [326,327].

Despite the ongoing quest to fully understand the molecular identity of the pore-forming unit, various models have been proposed. These range from models that involve regulatory proteins, such as cyclophilin D (Cyp-D), adenine nucleotide translocator (ANT), VDAC, hexokinase-II (HK-II), mitochondrial creatine kinase (mtCK), and translocator protein of 18 kDa (TSPO) [328,329], to more recent hypotheses that also encompass ATP synthase [329,330].

Specifically concerning ALS, high expression levels of mPTP regulators, including VDAC, ANT, and Cyp-D, were found in mouse MNs, as well as nitration of Cyp-D and ANT in spinal cord fractions of symptomatic ^G93Ahigh^-mSOD1 mice [331]. Additionally, the genetic deletion of Cyp-D delayed disease onset and extended the survival of ^G93A^-mSOD1 mice expressing high and low levels of the mutant protein. These findings suggest that mPTP is involved in the causal mechanisms of MN degeneration in mouse ALS, partly due to oxidative and nitrative damage to pore regulatory proteins [331]. Moreover, a study comparing knockout Cyp-D mice overexpressing SOD1 mutations (G93A, G85R, and G37R) with littermates overexpressing Cyp-D found that the absence of Cyp-D improved mitochondrial ATP synthesis, reduced the levels of misfolded SOD1 aggregates in the spinal cord, and suppressed MN death [332]. Intriguingly, an increased expression of Cyp-D was also observed in association with mitochondrial ROS production and enhanced mitoflash activity in the muscle of SOD1^G93A^ transgenic mice at the onset of symptoms [333]. Furthermore, the application of cyclosporin A (CsA), an inhibitor of Cyp-D, attenuated SOD1^G93A^-induced mitoflash activity, suggesting the involvement of mPTP opening in muscle dysfunction [333].

Considering that the denervation of skeletal muscle deprives it of any electric stimulation—resulting in the loss of physiological calcium uptake by mitochondria, increasing mitochondrial ROS production and triggering mPTP opening—it is plausible to propose that mPTP opening is a key element in the denervation process observed in ALS [334]. Further research is required to elucidate this aspect comprehensively.

### 2.9. Mitochondrial Dysfunction as Cause and/or Consequence in ALS

As described above, mitochondrial dysfunction in ALS is a multifaceted phenomenon, and determining whether it is a cause or consequence of the disease remains a challenge. While the literature suggests that mitochondrial impairment is observed early in the disease process (e.g., [335,336,337]), pinpointing the exact initiating event is complicated by the heterogeneity of ALS cases [187,338,339].

The fact that numerous studies reported mitochondrial abnormalities in ALS patients and animal models at early, pre-symptomatic stages suggests that mitochondrial dysfunction is an early event in the disease cascade, possibly contributing to the initiation or acceleration of ALS pathogenesis. On the other hand, mitochondrial dysfunction could also be a consequence of the disease’s evolution. Motor neuron degeneration, protein aggregation, and other ALS-related mechanisms may induce stress on mitochondria, leading to their dysfunction as a downstream effect [340]. Importantly, mitochondrial dysfunction and ALS pathogenesis may engage in a reciprocal relationship, forming a vicious cycle; initial mitochondrial impairment could exacerbate disease progression, while ongoing neurodegeneration may further compromise mitochondrial function. The presence of specific genetic mutations associated with both mitochondrial function and ALS adds another layer of complexity. Interactions between genetic susceptibility and environmental factors may influence the dynamics of mitochondrial involvement in ALS. The elusive nature of causality underscores the urgency of continued investigations to unravel the intricate relationship between mitochondria and ALS, paving the way for targeted and effective therapeutic interventions.

## 3. Preclinical and Clinical Endeavors Targeting Mitochondria

Understanding and harnessing ways to enhance mitochondrial function has emerged as a promising avenue for therapeutic development in ALS and other neurodegenerative diseases. This section delves into innovative approaches, compounds, and therapies that target mitochondria to restore their vitality, promote cellular energy production, and ultimately address the underlying causes of diseases associated with mitochondrial dysfunctions, such as ALS. From metabolic enhancers to antioxidants and mitochondria-targeted agents, these approaches offer hope for more effective treatments and improved quality of life for those affected by ALS and other mitochondrial-related conditions (Figure 3). The results of relevant clinical trials (Table 1) will also be discussed in this section.

### 3.1. Approaches for Direct Enhancement of Mitochondrial Function

Metabolic enhancers represent a compelling class of therapeutic interventions in the context of ALS and other neurodegenerative disorders. These approaches aim to boost cellular energy production and stimulate mitochondrial function. As metabolic deficits are a hallmark feature in ALS, emerging research suggests that enhancing metabolic pathways within mitochondria may hold the key to slowing or mitigating disease progression.

#### 3.1.1. Dichloroacetate

Dichloroacetate (DCA), investigated in multiple neurodegenerative diseases [341], represents an alternative approach for indirectly enhancing mitochondrial function. As a pyruvate dehydrogenase kinase (PDK) inhibitor, DCA stimulates the conversion of pyruvate to acetyl coenzyme A (AcCoA), supplying additional energy substrates to the TCA cycle (Figure 3A).

Administered to mutant SOD1^G93A^ or SOD1^G86R^ mice at 500 mg/kg/day during the pre-symptomatic stage, DCA improved survival, delayed disease onset, reduced spinal motor neuron loss, and enhanced mitochondrial function [173,342]. For example, DCA treatment in SOD1^G86R^ mice not only improved glycolytic capacity but also upregulated the expression of genes associated with mitochondrial biogenesis (e.g., *Pgc-1α* and *Mfn2*), believed to impede disease progression in this animal model [173]. Moreover, DCA (5 mM) improved the mitochondrial function of abnormal glial cells isolated from the spinal cords of adult paralytic SOD1^G93A^ rats, enhancing respiratory capacity and decreasing toxicity to MNs [343].

Additionally, administering DCA (100 mg/kg for 10 days) to symptomatic SOD1^G93A^ rats reduced MN degeneration, gliosis, and the number of GFAP/S100β double-labeled hypertrophic glial cells in the spinal cord. These findings indicate DCA’s potential therapeutic strategy for ALS by modulating glial metabolism and mitigating MN degeneration [343].

However, no clinical trials have assessed DCA effectiveness in ALS patients, necessitating further research.

#### 3.1.2. Ketogenic and High-Fat Diets

Various dietary interventions aim to delay ALS progression and counteract the hypermetabolism associated with the condition. Two notable approaches include the Ketogenic Diet (KD) [344,345] and the High-Fat Diet (HFD) [136,346,347,348] (Figure 3A).

A KD, characterized by high fat intake, increases circulating ketones, while restricting carbohydrates and proteins [349,350]. The primary ketones acetoacetate and D-β-3-hydroxybutyrate, produced in the liver, serve as an energy source when glucose availability is limited [351]. Studies report that a KD, comprising 60% fat, 20% carbohydrates, and 20% protein, improves motor function and safeguards MNs in SOD1^G93A^ mice [344,345].

This diet exerts its effects partially through alterations in mitochondrial function, promoting ATP synthesis, and restoring the activity of complex I in the ETC, which is often impaired in ALS [345]. Furthermore, it was demonstrated that caprylic triglyceride, a precursor to ketone bodies, enhanced motor function and protected MNs in SOD1^G93A^ mice by boosting oxidative metabolism, thereby increasing mitochondrial basal and maximum oxygen consumption [344]. Converted rapidly to caprylic acid, it easily traverses membranes, becoming β-oxidized to AcCoA in the mitochondria. Consequently, it supplies ketone bodies to the TCA cycle, serving as a rapid energy source when cellular glucose levels are low. Notably, neither the KD nor caprylic triglyceride significantly altered the survival of SOD1-^G93A^ transgenic mice [344,345].

Regarding the HFD, studies have indicated its potential to slow disease progression in ALS mouse models. One study employing a diet composed of 47% fat, 38% carbohydrates, and 15% protein reduced disease progression in a SOD1^G93A^ mouse model [348]. Similarly, another study utilizing a HFD comprising 21% fat and 0.15% cholesterol extended the mean survival of SOD1^G86R^ mice [136]. To investigate the impact of high-caloric diets on ALS patients, a double-blinded trial known as the LIPCAL-ALS study (NCT02306590) enrolled 201 patients; participants were assigned to receive either a high-caloric fatty diet (HCFD, 405 kcal/day, 100% fat) or a placebo in conjunction with riluzole (100 mg/day). However, the results did not provide conclusive evidence of a life-prolonging effect of the diet on the overall ALS patient population. Nevertheless, post-hoc analysis revealed a significant survival benefit for a subgroup of fast-progressing patients [352]. Given the potential influence of caloric content on the intervention’s efficacy, a clinical Phase I LIPCALII study (NCT04172792) is set to explore whether an ultra-high caloric diet (UHCD), featuring twice the caloric content compared to LIPCAL-ALS, can be well tolerated by ALS patients over four weeks and if it can exceed the beneficial effects of LIPCAL-ALS. In another study [353] assessing the effects of a high-caloric nutrition protocol on ALS patients with percutaneous gastrostomy, 40 patients were randomly assigned to either a routine diet (control group) or high-caloric nutrition combined with the routine diet (Ensure group) for six months. This study demonstrated a significant increase in cumulative survival rates in the Ensure group, suggesting the potential of this approach to enhance nutritional status and longevity.

In summary, the mechanisms through which HFD and KD modify ALS disease progression in mouse models are unclear. However, it is suggested that the high fat content of these diets might elevate phospholipids and cholesterol, which are crucial components for axonal membrane assembly and regeneration [354]. In alignment with this hypothesis, it was demonstrated that abnormally elevated cholesterol levels were associated with increased survival in ALS patients, by more than 12 months, suggesting hyperlipidemia as a prognostic factor [355].

#### 3.1.3. Acetyl-Carnitine

Acetyl-carnitine (ALC) is a crucial cellular source of acetyl groups (Figure 3A), particularly in high-energy demanding situations, playing a pivotal role in transporting long-chain fatty acids across mitochondrial membranes and limiting β-oxidation rates [356].

Studies administering 50 mg/kg/day of ALC orally before disease onset significantly delayed symptoms, slowed motor function deterioration, and extended lifespan in mutant SOD1^G93A^ mice; subcutaneous ALC injection to symptomatic mutant SOD1^G93A^ mice improved survival [357].

In a randomized double-blind, placebo-controlled Phase II trial involving 82 patients receiving 3 g/day of ACL or a placebo with riluzole (100 mg/day), ALC demonstrated mild enhancements in ALSFRS score and respiratory capacity. Notably, it doubled the median survival range compared to the placebo group, indicating effectiveness, tolerability, and safety in ALS treatment [358]. A recent study involving 32 ALS patients revealed ALC’s improvement in the redox state, persisting six months post-treatment, offering potential disease biomarkers and drug effects indicators in clinical practice and trials [359].

### 3.2. Antioxidants

Mitochondrial dysfunction, either causal or consequential to oxidative stress, contributes to ALS pathogenesis. While oxidative damage and mitochondrial dysfunction are recognized contributors to ALS pathogenesis, it is disheartening that all antioxidant treatments tested in patients have thus far proved ineffective. This highlights the urgent need for advancements in antioxidant therapies for ALS treatment. In the following section, we delve into various preclinical and clinical trials, both completed and ongoing, that explore antioxidants capable of influencing mitochondrial homeostasis (Figure 3B).

#### 3.2.1. N-acetyl-L-cysteine (NAC)

N-acetyl-L-cysteine (NAC), a membrane-permeable antioxidant, replenishes cellular cysteine and glutathione pools, mitigating free radical damage [360,361] (Figure 3B). In a preclinical study, NAC (1 mM, 24 h) attenuated mitochondrial ROS production, restored MTT reduction rates to control levels, and elevated ATP levels in human neuroblastoma SH-SY5Y cell lines harboring the SOD1^G93A^ mutation [362]. In SOD1^G93A^ transgenic mice, a daily dose of 2.0 mg/kg significantly extended survival and improved motor performance [360]. However, a double-blind, placebo-controlled clinical trial involving 110 ALS patients using subcutaneous NAC infusion (50 mg/kg daily) did not show a substantial increase in 12-month survival or disease progression slowdown [363]. In G93A mice, intranasal NAC administration with the nanocarrier PEG-PCL-Tat significantly increased spinal cord accumulation, extending median survival by 11.5 days. These findings highlight the potential of this approach as a promising Drug Delivery System for ALS therapeutics [364]. However, the potential benefits of NAC in ALS remain uncertain, necessitating further clinical trials in humans. The ability of NAC to traverse the Blood–Brain Barrier (BBB) has been a subject of controversy and is likely influenced by dosage and administration routes [365]. Consequently, multiple NAC derivatives have been synthesized to overcome this limitation. Important examples include N-acetylcysteine ethyl ester (NACET), which is proposed to improve pharmacokinetics but undergoes rapid transformation into NAC and cysteine resulting low plasma levels [366]; N-acetylcysteine butyl ester (NACBE), which is highly lipophilic and was shown to have superior effects after oxidative insult exposure compared to NAC [367]; and N-acetylcysteine amide (NACA), which was developed to enhance lipophilicity, membrane permeability, and the capability to traverse the BBB [368]. Previous studies have supported the protective properties of NACA, suggesting its potential clinical utility [369].

#### 3.2.2. Edaravone

Edaravone, the active component of Radicut^®^, is a potent free radical scavenger widely employed in the treatment of cerebral ischemia in Japan [370,371]. Its neuroprotective role arose from its ability to eliminate lipid peroxides and hydroxyl radicals [372,373] (Figure 3B). While the precise mechanisms are not fully understood, it has been proposed that in addition to its radical-scavenging properties, edaravone might inhibit the mPTP, contributing to its neuroprotective effects [374]. It benefits various CNS cell types, including neurons, microglia [375], astrocytes [376], and oligodendrocytes [377], partly attributable to its anti-inflammatory properties [378]. Preclinical studies indicate improved motor function, slowed disease progression, and mitigated motor neuron degeneration in transgenic SOD1 rodent models of ALS treated with edaravone doses ranging from 1.5 to 15 mg/kg [379,380].

In an open-label Phase II study with 20 ALS patients, intravenous administration of edaravone (30 or 60 mg daily) was safe, well-tolerated, and slowed disease progression, as measured by the ALS-FRS scale, during the six-month treatment period compared to the six months before edaravone administration [381]. A subsequent double-blind, placebo-controlled study with 102 ALS patients showed a smaller reduction in ALSFRS-R scores in the edaravone group over a 24-week treatment period [371].

In a recent Phase III study (NCT01492686), a 24-week, double-blind, parallel-group study of edaravone showed less decline in ALSFRS-R scores at 6 months and less deterioration in quality of life in patients receiving edaravone compared to those receiving standard care [382]. Currently, edaravone is approved for ALS treatment in Japan and South Korea and it was also approved by the FDA in May 2017 [383]. However, the precise mechanism of action of edaravone in ALS treatment remains to be fully elucidated.

A Phase I trial of an oral formulation of edaravone (TW001) developed by the Treeway company demonstrated safety, tolerability, and adequate exposure levels [384]. Furthermore, in a Phase III trial (NCT04165824), oral edaravone showed a favorable safety profile in ALS patients after 48 weeks of treatment [385]. A recent meta-analysis [386] suggests potential clinical benefits of edaravone in ALS treatment, with no significant increase in adverse events or deaths in compiled randomized clinical trials. However, more high-quality research is needed for further confirmation due to the small sample sizes in the included studies.

#### 3.2.3. Melatonin

Melatonin (N-acetyl-5-methoxytryptamine), a neurohormone secreted by the pineal gland, possesses ROS-scavenging activity and amphiphilic properties, permeating both lipophilic and hydrophilic cellular environments [387] (Figure 3B). Its potential as an experimental drug has been explored in various neurodegenerative diseases characterized by excessive ROS production owing to its robust antioxidant properties [388].

In addition to acting as a potent free radical scavenger, melatonin augments cellular antioxidant defenses by stimulating vital antioxidant enzymes (SOD, glutathione peroxidase, and glutathione reductase) and elevating GSH levels [389]. Furthermore, melatonin plays a role in preserving mitochondrial homeostasis, decreasing free radical generation, and protecting mitochondrial ATP synthesis by stimulating the activities of complexes I and IV [390].

In transgenic SOD1^G93A^ mice, the oral administration of melatonin (57–88 mg/kg/day) at the pre-symptomatic stage delayed disease progression and extended survival [87]. The same study demonstrated that the rectal administration of 300 mg/day of melatonin in 31 sALS patients was well tolerated over 2 years, reducing circulating serum protein carbonyl levels. However, it did not show upregulation of genes encoding antioxidant enzymes [87]. The decreased oxidative damage in ALS patients under melatonin treatment, coupled with its established safety in humans, emphasizes the need for further clinical trials to elucidate its neuroprotective effects in ALS.

More recently, it was reported that administering melatonin (30 mg/kg) to pre-symptomatic SOD1^G93A^-transgenic mice significantly delayed disease onset, neurological deterioration, and mortality [391]. These effects involved inhibition of the caspase-1/cytochrome c/caspase-3 pathways and the loss of melatonin receptor 1A. Conversely, melatonin administration (0.5, 2.5, and 50 mg/kg, i.p.) to pre-symptomatic SOD1^G93A^-transgenic mice reduced their survival [392]. This study revealed increased MN loss, elevated 4-hydroxynonenal (4-HNE) levels, and upregulated expression of human SOD1 in mice treated with melatonin compared to untreated animals. These findings suggest that melatonin might exacerbate the phenotype in the SOD1^G93A^ mouse ALS model due to the upregulation of toxic human SOD1, which may override its antioxidant and anti-apoptotic effects [392]. This raises the possibility that the SOD1^G93A^ ALS mouse model may not be ideal for evaluating the neuroprotective properties of melatonin or other compounds with complex antioxidative properties. Further research is warranted to unravel the mechanisms of melatonin’s action and determine whether its antioxidant and anti-apoptotic effects can translate into clinical benefits.

#### 3.2.4. Mitochondria-Targeted Antioxidants

One promising avenue in ALS research involves mitochondria-targeted antioxidants, such as 10-(60-ubiquinonyl) decyltriphenylphosphonium, known as MitoQ (Figure 3B). This compound features a triphenylphosphonium (TPP) functional group linked to the antioxidant ubiquinone [393] that penetrates biological membranes and selectively accumulates within mitochondria, driven by the ΔΨ_m_ [394]. Positioned within mitochondria, MitoQ effectively shields these critical organelles from oxidative damage [393,395].

Inside mitochondria, complex II reduces the ubiquinone moiety of MitoQ to its active ubiquinol form, bolstering the defense against oxidative damage [395]. Various oxidants can convert ubiquinol back to ubiquinone, efficiently reversed by the respiratory chain [396] and ensuring continual recycling. MitoQ efficiently mitigates oxidative damage in chronic hepatitis C virus patients, decreasing liver damage [397]. Additionally, it has shown promise in neurodegenerative diseases like Parkinson’s [398,399] and Alzheimer’s [400,401], by decreasing oxidative damage. However, a Phase II clinical trial for Parkinson’s disease (NCT00329056) yielded disappointing results [402].

Despite the setback in Parkinson’s disease trials, MitoQ has shown potential in ALS; it ameliorated nitroxidative stress and mitochondrial dysfunction in astrocytes expressing SOD1^G93A^, decreasing toxicity to MNs in co-cultures [403]. In SOD1^G93A^ mice, MitoQ (500 µM) administration improved the ALS phenotype by slowing mitochondrial function decline in the spinal cord and quadriceps muscle, increasing the lifespan of the animals [404]. This treatment reduced nitroxidative markers and pathological signs in the spinal cord along with the recovery of neuromuscular junctions and a marked increase in hindlimb strength [404]. MitoQ rapidly crosses the BBB [400] and is well tolerated in animals and humans [397], with nausea being the most common side effect [405].

These findings underscore the potential of mitochondria-directed antioxidants as a strategy to delay ALS symptoms, warranting further development.

Another mitochondria-targeted antioxidant in ALS research is the mitochondria-targeted carboxy-proxyl (Mito-CP). Like MitoQ, this features a TPP cation covalently coupled to carboxy-proxyl nitroxide, accumulating within mitochondria [406]. Low doses of Mito-CP (1–10 nM) effectively prevented MN death expressing SOD1^G93A^ induced by proapoptotic stimuli that trigger ROS formation [407]. It also prevented mitochondrial dysfunction, decreasing O_2_^.−^ production in SOD1^G93A^ astrocytes and promoting MN survival [403]. 

While these results are promising, further investigations are necessary to explore other mitochondriotropic compounds, focusing on reducing toxicity and enhancing therapeutic efficacy.

In addition, it is crucial to prioritize investment in the development of ALS models that faithfully represent the diverse subtypes of the disease. This approach aims to address the challenge of translating positive effects observed in preclinical trials with antioxidants into meaningful efficacy during clinical trials. These models should serve as robust platforms for conducting preclinical trials before advancing to clinical trials [22]. Human induced pluripotent stem cells have opened avenues to explore therapeutic development relevant to human diseases [408,409]. An example is the generation of MNs from a patient-derived iPSC line carrying the SOD1-A4V mutation that demonstrated significant disease phenotypes, including proteinopathy, structural attrition, axonopathy, synaptic pathology, and functional defects. This model holds the potential to emerge as a robust preclinical platform for evaluating the therapeutic efficacy of diverse molecules in addressing this disease [66].

### 3.3. Antiapoptotic Agents

#### 3.3.1. mPTP-Targeting Agents

As discussed in Section 2.8, emerging evidence points towards the involvement of mPTP in ALS pathogenesis. Initial studies used CsA, which, when administered intracerebroventricularly (25 µg every other day) at the symptomatic stage in SOD1^G93A^ mice, extended their survival [410]. Similarly, intracerebroventricular administration of CsA (20 µg/mouse/week) in pre-symptomatic SOD1^G93A^ mice delayed the onset of hindlimb weakness, prolonged the time from onset to paralysis, and extended life span [411]. However, it is important to note that CyA struggles to cross the BBB [411,412] and its benefits could be confounded by its immunosuppressant effects due to calcineurin inhibition. Therefore, it is advisable to explore other mPTP inhibitors that lack calcineurin effects.

One such strategy involves mPTP stabilization using cholest-4-en-3-one oxime, known as olesoxime (TRO19622), which binds to VDAC and TSPO [413,414] (Figure 3C). Subcutaneous administration of TRO19622 (3 or 30 mg/kg) in pre-symptomatic SOD1^G93A^ mice improved motor function and prolonged survival [415]. Additionally, administration of TRO19622 (600 mg/kg of food pellets) in pre-symptomatic SOD1^G93A^ mice delayed muscle denervation, decreased astrogliosis, prevented microglia activation, and protected MNs in the lumbar spinal cord, suggesting its potential as a neuroprotective agent to delay ALS neurodegeneration [416]. However, a Phase II–III clinical trial failed to demonstrate efficacy in 512 ALS patients receiving 330 mg TRO19622 daily compared to a matching placebo group receiving 50 mg riluzole twice a day for 18 months. None of the assessed parameters, including slow vital capacity, manual muscle testing, and rates of deterioration of ALSFRS-R scores, revealed a significant clinical benefit of TRO19622 treatment compared with the placebo, except for a minimal increase in the ALSFRS-R global score over 9 months [417]. Several factors may explain the absence of clinical efficacy, including differences in the timing of administration relative to disease onset, the limitations of animal models in predicting clinical outcomes [418], and the need for further investigation into the survival-promoting effects of TRO19622 on human MNs derived from ALS patient induced pluripotent stem cells [419].

Another compound of interest is GNX-4728, a cinnamic anilide derivative that inhibits mPTP opening and has shown promise in the SOD^G93A^ mouse model (Figure 3C). Systemic treatment with GNX-4728 (15 mg/kg) in C57BL/6 mice significantly increased mitochondrial calcium retention capacity (CRC) in the heart and brain. Importantly, the increase in CRC in the brain suggests its potential to cross the BBB [420]. Furthermore, systemic administration of GNX-4728 (300 µg every other day, i.p.) in pre-symptomatic SOD1^G37R^ mice delayed disease onset, increased lifespan, protected against motor neuron and mitochondrial degeneration, attenuated spinal cord inflammation, and preserved neuromuscular junction (NMJ) innervation in the diaphragm in ALS mice [420].

These findings highlight the potential of mPTP-targeting agents as a therapeutic approach in ALS, with the need for further research to optimize their clinical translation.

#### 3.3.2. Rasagiline

Rasagiline, a monoamine oxidase B inhibitor primarily used to treat Parkinson’s disease [421,422], has shown neuroprotective properties beyond its MAO inhibitory activity [423] (Figure 3C). In vitro studies suggest its neuroprotective role, partly mediated by anti-apoptotic properties [424,425,426]. Rasagiline may protect mitochondria by preventing mPTP opening, inhibiting caspase 3 activation [425], or by upregulating the anti-apoptotic Bcl-2 and Bcl-xL genes and downregulating the pro-apoptotic Bad and Bax genes [426].

In ALS, oral rasagiline (0.5–2 mg/kg/day), administered alone or with riluzole (30 mg/kg/day) at the pre-symptomatic stage, improved motor performance and extended survival in SOD1^G93A^ mice [427]. A Phase II clinical trial involving 23 ALS patients receiving rasagiline (2 mg/day) showed increased ΔΨ_m_ and oxygen radical antioxidant capacity (ORAC) in lymphocytes and a decrease in the apoptotic marker Bcl-2/Bax ratio [428]. While this trial did not find significant improvements in the rate of ALSFRS-R score decline, it did identify differences in symptom duration among patients administered who received rasagiline compared to placebo controls (NCT01232738) [428]. This raises the question of whether rasagiline-induced mitochondrial changes could slow motor function decline and extend ALS patient survival.

However, another Phase II clinical trial involving 80 ALS patients receiving rasagiline (2 mg/day for 12 months) did not demonstrate improvements in mitochondrial and molecular biomarkers (ΔΨ_m_, ORAC, and Bcl-2/Bax ratio) or differences in the average 12-month ALSFRS-R slope between the rasagiline and placebo groups. Variability among patients and differences in sample processing timing at various centers involved in this clinical trial (NCT01786603) may explain the discrepant results regarding rasagiline’s impact on mitochondrial markers [429].

Post-hoc analysis of a Phase II clinical trial involving 252 patients (NCT01879241) suggested that 1 mg of rasagiline, in addition to riluzole (100 mg/day), could slow disease progression in patients with normal to fast disease progression. This effect was observed in function (ALSFRS-R decline at 6, 12, and 18 months) and survival (at 6 and 12 months) [430]. This study emphasizes the importance of stratifying disease progression rates in clinical trials, as the response to drugs and their disease-modifying effects may vary according to progression rate [430]. However, as the trial did not initially specify an analysis by progression rate, cautious interpretation is necessary. Future clinical trials are needed to establish rasagiline’s therapeutic potential in ALS, especially for normal- to fast-progressing forms.

In conclusion, rasagiline’s effects in ALS research reveal its complex potential as a therapeutic agent, with varying outcomes in clinical trials. Further research and careful patient stratification are essential to unlock its full therapeutic benefits in ALS treatment.

## 4. Conclusions

Throughout this review, we have delved into the complex interplay between mitochondrial dysfunction and the pathogenesis of ALS, a multifaceted disease characterized by the progressive degeneration of MNs and with elusive therapeutic options.

Mitochondria have emerged as a compelling convergent target for ALS treatment. Mitochondrial dysfunction, encompassing bioenergetic deficits, oxidative stress, impaired calcium handling, and disrupted dynamics, plays a pivotal role in the cascade of events that ultimately leads to motor neuron demise in ALS. The evidence gleaned from preclinical models and clinical trials underscores the significance of mitochondrial abnormalities in this devastating disease. As the intricate web of mechanisms underpinning ALS has unraveled, mitochondria have emerged as a nexus where various pathological processes converge. The potential of mitochondria as a therapeutic target in ALS is multifaceted and encompasses:Restoring Energy Balance: Mitochondrial dysfunction in ALS often leads to a decline in cellular energy production, which is particularly detrimental to highly energy-demanding MNs. Strategies aimed at rejuvenating mitochondrial function and ensuring an adequate supply of ATP hold promise in sustaining motor neuron health and function.Attenuating Oxidative Stress: Mitochondria are both sources and targets of oxidative stress in ALS. Therapies aimed at reducing mitochondrial-generated ROS and bolstering endogenous antioxidant defenses offer potential avenues for alleviating oxidative damage.Regulating Calcium Homeostasis: Impaired calcium homeostasis in ALS contributes to mitochondrial dysfunction and subsequent neuronal death. Interventions that restore the calcium balance within neurons and mitochondria may mitigate excitotoxicity and improve neuronal survival.Enhancing Mitochondrial Dynamics: Maintaining a healthy mitochondrial network through processes such as fission, fusion, and mitophagy is crucial for cellular health. Targeting these dynamic processes to ensure the timely removal of damaged mitochondria and the efficient distribution of healthy ones may have therapeutic implications.Unveiling Genetic Insights: Genetic mutations associated with ALS, such as those in the *SOD1*, *C9ORF72*, and *FUS* genes, can directly impact mitochondrial function. Understanding the precise mechanisms by which these mutations affect mitochondria could pave the way for gene-specific therapies.

As we reflect on the current state of ALS research and treatment, it is evident that mitochondrial-targeted strategies hold significant potential. Several promising compounds, such as NAC, CoQ10, MitoQ, and other mitochondria-targeting antioxidants, have shown efficacy in preclinical models by mitigating mitochondrial dysfunction and delaying disease progression. These candidates provide a glimmer of hope for ALS patients and their families. 

## Figures and Tables

**Figure 1 cells-13-00248-f001:**
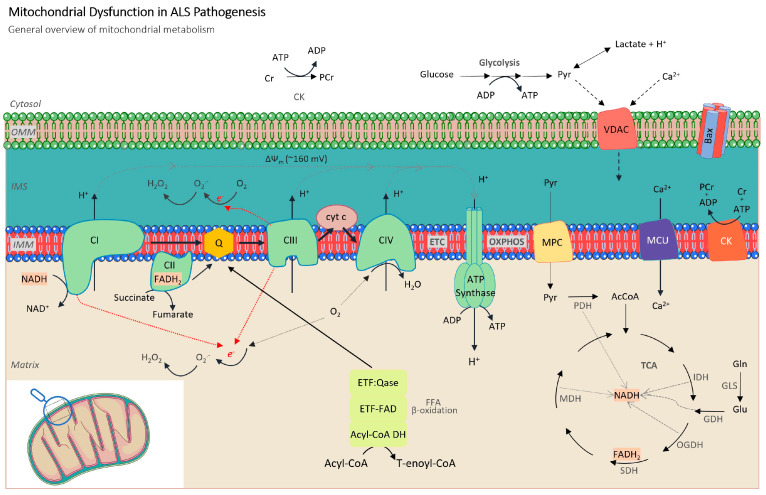
Overview of mitochondrial functions. Mitochondrial bioenergetics is driven by the oxidation of different substrates and is stimulated by calcium. Flux of electrons through the electron transport chain creates a transmembrane proton gradient of about 160 mV in the resting state, which fuels Adenosine Triphosphate (ATP) synthesis in the mitochondrial matrix. Leakage of electrons in some bioenergetic reactions generates reactive oxygen species (ROS) that are involved in important cellular signaling processes. Abbreviations: ADP: Adenosine Diphosphate; CI: Complex I; CII: Complex II; CIII: Complex III; CIV: Complex IV; Cyt c: Cytochrome c; ETF: Electron Transfer Flavoprotein; FAD: Flavin Adenine Dinucleotide; IMM: Inner Mitochondrial Membrane; MCU: Mitochondrial Calcium Uniporter; MPC: Mitochondrial Pyruvate Carrier; ΔΨ_m_: Mitochondrial Transmembrane Electrochemical Potential; NAD: β-Nicotinamide adenine dinucleotide; NADH: β-Nicotinamide adenine dinucleotide 2′-phosphate reduced form.

**Figure 2 cells-13-00248-f002:**
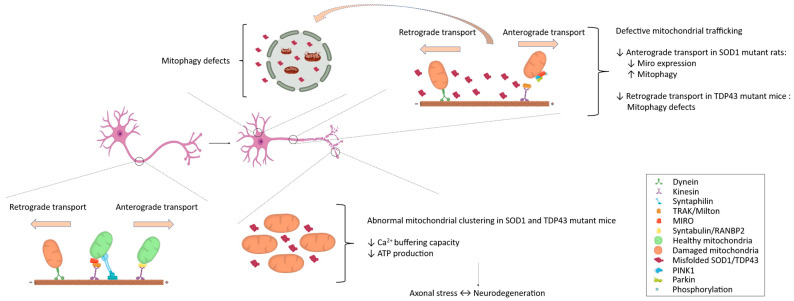
Dysregulated mitochondrial trafficking in ALS Models. Anomalies in mitochondrial trafficking are a prevalent phenomenon across various ALS models. Notably, both anterograde and retrograde transport processes are compromised in this context. Intriguingly, emerging research suggests a temporal sequence in which retrograde transport disruption precedes anterograde transport impairment. Dysfunction in retrograde transport is intimately associated with mitophagy deficits, resulting in the inefficient removal of dysfunctional mitochondria in TDP 43 ALS mutant mice. Conversely, cortical neurons from the E18 embryos of ALS SOD1 mutant rats exhibit alterations in anterograde transport linked to a reduction in Miro expression, ultimately leading to mitochondrial transport stagnation. These perturbations in both transport pathways culminate in abnormal mitochondrial clustering, diminished Ca^2+^ buffering capacity, and compromised ATP production. In extreme cases, these abnormalities can trigger axonal stress, ultimately contributing to cell death and neurodegeneration.

**Figure 3 cells-13-00248-f003:**
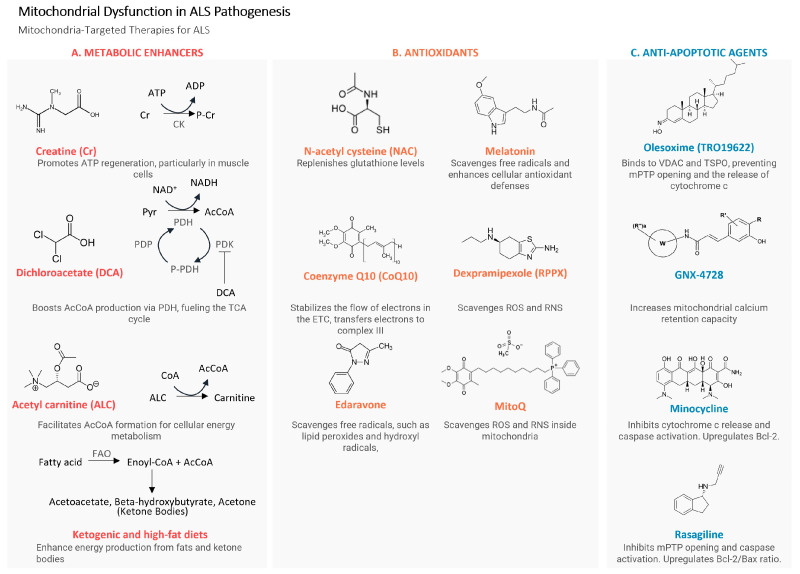
Mitochondria-targeted therapies for ALS featuring (**A**) metabolic enhancers, (**B**) antioxidants, and (**C**) anti-apoptotic agents. Abbreviations: AcCoA: acetyl coenzyme A; CK: Creatine kinase; ETC: electron transport chain; FAO: fatty acid beta-oxidation; mPTP: mitochondrial permeability transition pore; PDH, pyruvate dehydrogenase; PDK, pyruvate dehydrogenase kinase; PDP, pyruvate dehydrogenase phosphatase; ROS: reactive oxygen species; RNS: reactive nitrogen species; TCA: tricarboxylic acid cycle.

**Table 1 cells-13-00248-t001:** Clinical trials discussed in this review.

Clinical Trial ID	Study Name	Condition	Phase/Type	Sponsor	Starting Date	End Date	Number of Patients
NCT01257581	Phase 2 Selection Trial of High Dosage Creatine and Two Dosages of Tamoxifen in Amyotrophic Lateral Sclerosis (ALS)	Amyotrophic Lateral Sclerosis (ALS)	Phase 2	Nazem Atassi, Massachusetts General Hospital	2011-03	2013-02	60
NCT02306590	Efficacy, Safety and Tolerability of High Lipid and Calorie Supplementation in Amyotrophic Lateral Sclerosis	Amyotrophic Lateral Sclerosis (ALS)	Prospective, multicenter, randomized, stratified, parallel-group, double-blind trial	Albert Christian Ludolph, Prof., University of Ulm	2015-02	2018-09	207
NCT04172792	Safety and Tolerability of Fat-rich vs. Carbohydrate-rich High-caloric Food Supplements in Patients with Amyotrophic Lateral Sclerosis (ALS)	Amyotrophic Lateral Sclerosis (ALS)	Phase 1	Albert Christian Ludolph, Prof., University of Ulm	2019-11	2021-04	64
NCT00243932	Clinical Trial of High Dose CoQ10 in ALS	Amyotrophic Lateral Sclerosis (ALS)	Phase 2	Columbia University	2005-04	2008-03	185
NCT01492686	Efficacy and Safety Study of MCI-186 for Treatment of Patients with Amyotrophic Lateral Sclerosis (ALS) 2	Amyotrophic Lateral Sclerosis (ALS)	Phase 3	Mitsubishi Tanabe Pharma Corporation	2011-12	2014-10	137
NCT04165824	Safety Study of Oral Edaravone Administered in Subjects with ALS	Amyotrophic Lateral Sclerosis (ALS)	Phase 3	Mitsubishi Tanabe Pharma America Inc.	2019-11	2021-10	185
NCT01281189	A Randomized, Double-Blind, Placebo-Controlled, Multi-Center Study of the Safety and Efficacy of Dexpramipexole in Subjects with Amyotrophic Lateral Sclerosis	Amyotrophic Lateral Sclerosis (ALS)	Phase 3	Knopp Biosciences	2011-03	2012-11	942
NCT00329056	A Double-Blind, Prospective, Randomized Comparison of 2 Doses of MitoQ and Placebo for the Treatment of Patients with Parkinson’s Disease	Parkinson’s Disease (PD)	Double-Blind, Prospective, Randomized Comparison	Antipodean Pharmaceuticals, Inc.	2006-05	2007-11	128
NCT00047723	Minocycline to Treat Amyotrophic Lateral Sclerosis	Amyotrophic Lateral Sclerosis (ALS)	Phase 3	National Institute of Neurological Disorders and Stroke (NINDS)	2003-01	2007-01	400
NCT01232738	A Multi-Center Controlled Screening Trial of Safety and Efficacy of Rasagiline in Subjects with Amyotrophic Lateral Sclerosis (ALS)	Amyotrophic Lateral Sclerosis (ALS)	Phase 2	Yunxia Wang, MD, University of Kansas Medical Center	2011-12	2013-05	36
NCT01786603	Rasagiline in Subjects with Amyotrophic Lateral Sclerosis (ALS)	Amyotrophic Lateral Sclerosis (ALS)	Phase 2	Richard Barohn, MD, University of Kansas Medical Center	2013-11	2016-07	80
NCT01879241	Efficacy, Safety, and Tolerability Study of 1 mg Rasagiline in Patients with Amyotrophic Lateral Sclerosis (ALS) Receiving Standard Therapy (Riluzole)-An AMG Trial with a Market Authorized Substance	Amyotrophic Lateral Sclerosis (ALS)	Phase 2	Albert Christian Ludolph, Prof., University of Ulm	2013-06	2016-08	252

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
