# Peer review of "Mitochondria: A Promising Convergent Target for the Treatment of Amyotrophic Lateral Sclerosis"

_cells, 2024, doi:10.3390/cells13030248_

Round 1

Reviewer 1 Report

Comments and Suggestions for Authors

This review describes what is known about the functional involvement of mitochondria in ALS and potential therapeutic avenues. In ALS, the upper motor neurons in the cerebral cortex and lower motor neurons die off, resulting in respiratory failure and death. A variety of stress pathways converge in ALS, including ER stress, inflammation and mitochondrial dysfunction. The latter is particularly associated with ALS already in early stages and evidenced by the involvement of many mitochondria-regulatory proteins in ALS. Their cumulative effect reduces OXPHOS activity, while increasing ROS production. Even ALS gene products not directly associated with mitochondria create dysfunction of this organelle. A source of mitochondria dysfunction may originate from lower glutathione or redox-preserving transcription factors, such as Nrf2. Accompanying these mitochondrial defects, often increased glycolysis is found in ALS patient tissue and cells. 

This is a very thoughtful and comprehensive review that requires little improvement. I was a delight to read!

Specific Points:

1.    I agree with the authors that mitochondria are central in ALS and that understanding their involvement will be key in providing treatment. However, given the bewildering array of proteins involved, could the authors speculate whether mitochondria are cause or consequence of the disease? 

2.    Mitochondrial fission and fusion are under redox control, see for example PMID 22945481 and 29924999. This important point is missing from the review and should be put into context with other oxidative stress readouts. 

3.    The availability of antioxidants across the blood-brain barrier should be discussed, especially for NAC. 

4.    The involvement of calcium in ALS is not yet well fleshed out. At the moment, the reader is left puzzled how calcium transfer into mitochondria could be altered in an ALS scenario. The authors should describe the basic calcium transfer mechanisms at ER-mitochondria contacts, especially within axons, and their potential changes upon increased oxidative stress. This topic involves oxidative modifications of the organellar interface and the calcium-handling proteins. In this context, AD is known to exhibit changes of this structure and the authors should point readers in that direction. 

Author Response

We want to express our sincere gratitude to both reviewers for your thoughtful and constructive feedback on our manuscript. Your insightful comments have significantly contributed to enhancing the quality and clarity of our paper. It has been invaluable in refining our arguments, addressing potential limitations, and ensuring the accuracy of the information presented. We truly appreciate the time and effort dedicated to evaluating our work and committed ourselves to incorporating their suggestions diligently. Thank you for your valuable contributions. We believe that your feedback significantly strengthened the revised version of our paper. 

Point-by-point responses to reviewer 1:

1- Mitochondrial dysfunction in ALS is a multifaceted phenomenon, and determining whether it is a cause or consequence of the disease remains a challenge. While the literature suggests that mitochondrial impairment is observed early in the disease process, pinpointing the exact initiating event is complicated by the heterogeneity of ALS cases.

Numerous studies have reported mitochondrial abnormalities in ALS patients and animal models at early, pre-symptomatic stages. These findings suggest that mitochondrial dysfunction is an early event in the disease cascade, possibly contributing to the initiation or acceleration of ALS pathogenesis. On the other hand, mitochondrial dysfunction could also be a consequence of the disease's evolution. Motor neuron degeneration, protein aggregation, and other ALS-related mechanisms may induce stress on mitochondria, leading to their dysfunction as a downstream effect. Importantly, mitochondrial dysfunction and ALS pathogenesis may engage in a reciprocal relationship, forming a vicious cycle. Initial mitochondrial impairment could exacerbate disease progression, while ongoing neurodegeneration may further compromise mitochondrial function. The presence of specific genetic mutations associated with both mitochondrial function and ALS adds another layer of complexity. Interactions between genetic susceptibility and environmental factors may influence the dynamics of mitochondrial involvement in ALS. In our review, we aim to present the current state of knowledge on mitochondria in ALS comprehensively. While we discuss the association between mitochondrial dysfunction and ALS, we acknowledge the need for further research to decipher the causative elements and temporal sequence of events. The elusive nature of causality underscores the urgency of continued investigations to unravel the intricate relationship between mitochondria and ALS, paving the way for targeted and effective therapeutic interventions. A new section discussing these topics was added in page 20, lines 959-980 of the revised manuscript.

2- It is undeniably true that mitochondrial dynamics are under redox control and, as we now discuss in the  revised manuscript (page 13, lines 596-611) this idea can actually help to explain the biphasic pattern that suggest an initial protective response involving mitochondrial fusion, followed by a gradual shift towards increased fission in different ALS models. On one hand, an oxidizing environment rich in GSSG strongly induces mitochondrial fusion by inducing the formation of disulfide mediated Mfn oligomers in a process requiring GTP hydrolysis. On the other hand, the depletion of the mitochondrial-located protein disulfide isomerase A1, an oxidoreductase with thiol reductase activity for Drp1, induces a ROS-mediated protein post-translational modification that ultimately leads to mitochondrial fragmentation, further ROS increase and mitochondrial respiratory impairment. 

3-  According to the suggestion, this revised version of the manuscript incorporates the contentious issue regarding the capability of N-acetylcysteine (NAC) to traverse the Blood-Brain Barrier (BBB). Additionally, it acknowledges the synthesis of multiple derivatives of NAC as an effort to overcome this limitation, pages 24-25, lines 1104-1114.

4- In this revised version, a novel section titled "2.6 Mitochondria and endoplasmic reticulum crosstalk" (pages 16-17, lines 767-819) was included. This section describes the role of endoplasmic reticulum-mitochondria associated membranes (MAMs) in maintaining calcium homeostasis. It emphasizes that certain ALS-associated mutations, such as those in TDP-43, FUS, and VAPB genes, are correlated with compromised enzymatic activities within MAM domains, and that particularly these mutations impact the transfer of calcium between the endoplasmic reticulum (ER) and mitochondria. These findings are further supported by a study showing that disruption of VAPB-PTPIP51 tethering, and related delivery of Ca2+ from ER stores to mitochondria, occurs in neurons derived from iPS cells obtained from patients carrying pathogenic c9orf72 expansions associated with ALS/FTD, as well as a study by Pilotto and colleagues, who discovered that ER stress in human C9ORF72-ALS/FTD iPSCs and C9orf72 mouse neurons initiates a compensatory mechanism, that involves the upregulation of GRP75 expression at MAMs, effectively counteracting early mitochondrial Ca2+ imbalance. Over time, PolyGA inclusions sequester GRP75, resulting in its diminished presence at the MAMs, leading to functional loss and subsequent mitochondrial dysfunction. This new section also outlines various molecules that have been under investigation as potential therapeutic agents addressing dysfunction at MAMs in ALS.

Reviewer 2 Report

Comments and Suggestions for Authors

Cunha-Oliveira and colleagues have contributed a review manuscript on mitochondria in ALS. The authors give an overview on ALS identifying the key issue that there is no cure for ALS. They segue into the theme of mitochondrial dysfunction in the different forms of ALS (sporadic and familial), including aberrant respiration and ATP production, oxidative stress, metabolic dysregulation, mitochondrial dynamics and biogenesis, mitophagy, mitochondrial trafficking, apoptotic mechanisms, and then preclinical and clinical pharmacological endeavors that have some relationships to mitochondria.

The review is distinguished by its topic coverage. It is content dense. There are 422 references. It has 1 figure and 1 table.

The review has some weaknesses that the authors should consider.

11.       As stated in line 99 the authors seek “to synthesize recent insight into the role of mitochondria and mitochondrial-mediated mechanisms in driving cellular damage.” However, with synthesis comes hopefully a new product, and, despite the reviews’ length, it is not clear what is new in this review and whether it leads to new ideas and critical research about mechanisms of disease in ALS and its treatment.  The authors need to make crystal clear that what they are trying to say is new because there are so many published reviews on mitochondria in ALS. Otherwise, this review is incrementally derivative.

22.       Despite the 422 references, the citations in this review are not quite fitting all the time. For example, reference 38 is used for work that reports on mitochondria in muscle samples in SOD1 mice. It does not.  Also, in some instances the original seminal papers are not cited but should be. Please reference proper papers.

33.       The description of interactions of mutant SOD1 with key mitochondrial proteins (line 118) should be more complete by including lysyl-tRNA synthetase (KARS). Also please be specific that the VDAC interaction with mutant SOD1 is with VDAC1.

44.       Despite the length of the review, it does not include contemporary work on modeling ALS using human induced-pluripotent stem cells (iPSCs) with directed differentiation to motor neurons. Reviewing elements of iPSC research is very important because it is genetically physiological and human relevant. Most transgenic mouse systems of ALS are non-physiological with high excess of foreign mutant protein (human) in a background of normal endogenous wildtype (mouse) protein. Mechanistically this can be difficult to interpret. This is particularly true for most SOD1 transgenic mice where many mitochondrial abnormalities (morphological, biochemical, and functional) have been reported. Generally, the results are impairments. Interestingly, human motor neurons with one mutant allele of SOD1 generally have normal mitochondrial morphology and normal to increased ATP production Frontiers | Human Motor Neurons With SOD1-G93A Mutation Generated From CRISPR/Cas9 Gene-Edited iPSCs Develop Pathological Features of Amyotrophic Lateral Sclerosis (frontiersin.org). Such contrary findings should be discussed briefly.

55.       The authors have a section (2.2) on “Role of Oxidative Stress Mechanisms.” This is good. The authors should cite Beckmans’ original work ALS, SOD and peroxynitrite | Nature. Also, the nitration of CyPD and ANT is relevant to point out here The mitochondrial permeability transition pore in motor neurons: involvement in the pathobiology of ALS mice - PubMed (nih.gov).

66.       In lines 311-312 the authors cite work regarding SOD1 aggregation using transfected mouse NSC34 cells. This is a questionable motor neuron cell. More relevant is work on human iPSC-derived motor neurons.   

77.       In lines 333-350, the authors appropriately discuss work on the role of NADPH oxidase in ALS. This discussion should be more balanced by the Trumbull KA et al (2012) work which casts some doubt on the role of NADPH oxidase in the SOD1 mouse model of ALS.

88.       The authors have a section (2.3) on metabolic dysregulation. This section is very important and should be emphasized because of the holistic thinking on the ALS disease process, particularly involving skeletal muscle. Citing the key work of Dupuis and Loeffler is great. The authors should also consider citing Frontiers | Skeletal Muscle-Restricted Expression of Human SOD1 in Transgenic Mice Causes a Fatal ALS-Like Syndrome (frontiersin.org) because it is also holistic and implicates adipose tissue too.

99.       The mitochondrial trafficking section might be better with the mitochondrial dynamics section, or after it.

110.   In the apoptotic mechanisms section (2.7) the authors should cite an original earlier paper (Neuronal death in amyotrophic lateral sclerosis is apoptosis: possible contribution of a programmed cell death mechanism - PubMed (nih.gov)) in addition to the later appearing supporting work.

111.   Section 3 (preclinical and clinical endeavors targeting mitochondria) is excessively long and should be reduced in length by at one half. This section should be reduced in length dramatically because much of it is old news and there has been already discussion of the 20 years of translation failure ALS Clinical Trials Review: 20 Years of Failure. Are We Any Closer to Registering a New Treatment? - PubMed (nih.gov) on many of the drugs discussed here.  Moreover, your table nicely summarizes the clinical trials. The discussion of creatine, Co-q10 dexpramipexole, and minocycline should be minimized or just referenced in the Table.

Comments on the Quality of English Language

The quality of English is appropriate.

Author Response

We want to express our sincere gratitude to both reviewers for your thoughtful and constructive feedback on our manuscript. Your insightful comments have significantly contributed to enhancing the quality and clarity of our paper. It has been invaluable in refining our arguments, addressing potential limitations, and ensuring the accuracy of the information presented. We truly appreciate the time and effort dedicated to evaluating our work and committed ourselves to incorporating their suggestions diligently. Thank you for your valuable contributions. We believe that your feedback significantly strengthened the revised version of our paper. 

Point-by-point responses to reviewer 2:

1- While we recognize the abundance of reviews on mitochondria in ALS, our manuscript distinguishes itself by providing a comprehensive review that spans diverse mitochondrial aspects implicated in ALS pathogenesis. We go beyond a simple compilation of existing information, aiming to weave together various threads of mitochondrial involvement to offer a holistic view, which is not present in other reviews. Our review systematically integrates findings from preclinical studies, clinical trials, and mechanistic insights, bridging the gap between basic research and potential therapeutic applications. This approach allows us to explore not only the current understanding of mitochondrial dysfunction in ALS but also the translational implications for future treatment strategies. One of the primary focuses of our review is the identification of mitochondria as a promising convergent target for ALS treatment. By presenting mitochondria-targeted therapies and their diverse mechanisms of action, we aim to guide future research and therapeutic development in a more targeted and effective manner. We critically evaluate the existing literature, highlighting gaps in knowledge and proposing future research directions. Our review aims to stimulate new ideas and critical thinking in the field, encouraging researchers to explore innovative avenues for understanding and treating ALS.

A unique aspect of our review is the compilation of various clinical trials investigating mitochondria-targeted therapies for ALS. This provides readers with a consolidated overview of the current landscape of clinical research in this area. In conclusion, our review offers a nuanced and integrated perspective on mitochondria in ALS, going beyond a mere recapitulation of existing information. The synthesis of diverse findings, identification of convergent targets, critical analysis, and compilation of clinical trials collectively contribute to the novelty and significance of our work. 

2- Thank you for noticing this. The entire text was re-read and the original seminal papers were included in the missing sites. The reference 38 was corrected in Section 2. Mitochondrial dysfunctions in ALS, Page 3, line 113

3- Thank you for your suggestion, the inclusion of lysyl-trna synthetase (Kars) and and the specification of the VDAC isoform (VDAC1) is extremely important in the description of SOD1 mutant interactions. In this revised manuscript these two points were reviewed, as observed in Section 2. Mitochondrial dysfunctions in ALS, Page 3, line 118-119.

4- Thank you for your relevant suggestion. In this revised manuscript, we referred to the fact that the transgenic model is a non-physiological model that may overestimate the mitochondrial alterations and need to be carefully interpreted, in Section 2.1 Alterations on mitochondrial respiration and ATP production, Page 5, lines 203-210 “It is crucial to consider that the SOD1 transgenic animal model features a high copy number of human mutSOD1, rendering it a non-physiological model that may lack many of the phenotypic alterations observed in ALS patients (PMID: 26344214, PMID: 20184514). The prevalence of studies documenting disturbances in mitochondrial respiration in SOD1 transgenic animal models, in contrast to the absence of impairment in mitochondrial respiration observed in studies involving SOD1A4V or SOD1G93A human MNs (PMID: 33328898), suggests that mitochondrial dysfunctions observed in the SOD1 transgenic animal model may be overestimated and should be carefully interpreted.

The importance of developing accurate cellular models for ALS, which could function as robust platforms for conducting preclinical trials prior to advancing to clinical trials, was also highlighted in section 3.2.4. Mitochondria-targeted antioxidants, page 26, lines 1218-1229. ”In addition, it is crucial to prioritize investment in the development of ALS models that faithfully represent the diverse subtypes of the disease. This approach aims to address the challenge of translating positive effects observed in pre-clinical trials with antioxidants into meaningful efficacy during clinical trials. These models should serve as robust platforms for conducting preclinical trials before advancing to clinical trials (PMID: 33274002). Human induced pluripotent stem cells have opened up avenues to explore therapeutic development relevant to human diseases (PMID: 23736002; PMID: 26416678). An example is the generation of MNs from a patient-derived iPSC line carrying the SOD1-A4V mutation that demonstrated significant disease phenotypes, including proteinopathy, structural attrition, axonopathy, synaptic pathology, and functional defects. This model holds the potential to emerge as a robust preclinical platform for evaluating the therapeutic efficacy of diverse molecules in addressing this disease (PMID: 33328898).

5- The original work by Beckman and colleagues on ALS, SOD and peroxynitrite (Nature) was included in Section 2.2 Role of oxidative stress mechanisms, Page 7, line 312 of the revised manuscript. The importance of nitration of Cyp-D and ANT and their relevance on the induction of mitochondrial permeability transition is reported in section 2.8 Apoptotic mechanisms, page 20, lines 940-944.

6- In this revised manuscript, work on SOD1 aggregation using iPSC-derived MNs was also included, highlighting its significance in understanding ALS disease progression, in Section 2.2 Role of oxidative stress mechanisms, page 7, lines 314-316

7- Thank you for pointing out this valuable aspect. We agree that a more balanced discussion is crucial for presenting a comprehensive view of the role of NADPH oxidase in ALS. We revised the manuscript to include a more nuanced discussion that incorporates the findings of Trumbull et al. 2012 (https://doi.org/10.1016/j.nbd.2011.07.015).This study challenged the previous notion that NADPH oxidase inhibition, particularly with apocynin and its derivative diapocynin, significantly extends the survival of SOD1G93A ALS mice. Their experiments, conducted with meticulous care and repeated trials at different institutions, failed to replicate the remarkable survival extension reported by Harraz et al. (2008) (https://doi.org/10.1172/JCI34060). The lack of consistent and significant protective effects in various trials raises important questions about the translational potential of NADPH oxidase inhibition as a therapeutic strategy for ALS.

We believe that incorporating these findings into our review provides a more balanced perspective, acknowledging both studies that suggest a role for NADPH oxidase and those that question its efficacy in the SOD1G93A mouse model of ALS. We emphasized the importance of considering diverse experimental outcomes in shaping our understanding of potential therapeutic targets in ALS, page 8, lines 358-366.

8- To underscore the holistic nature of ALS, we included the study conducted by Martin and Wong that provides evidence of a relationship between muscle and adipose tissue within the context of this disease (page 10, lines 442-446).

9- As suggested, the mitochondrial trafficking section (2.5) was placed after the mitochondrial dynamics section (2.4), pages 14-16.

10- The article "Neuronal death in amyotrophic lateral sclerosis is apoptosis: possible contribution of a programmed cell death mechanism" was cited in Section 2.8, Page 19, line 910 of the revised manuscript

11- To improve the readability of Section 3 Preclinical and Clinical Endeavors Targeting Mitochondria (pages 20-28, lines 981-1306), the text was summarized and details about trials involving creatine, CoQ10, dexpramipexole, and minocycline were excluded from the main text. Instead, the studies of  creatine, CoQ10, dexpramipexole, and minocycline are only referenced in Table 1, facilitating the access to the relevant information.

Round 2

Reviewer 2 Report

Comments and Suggestions for Authors

The authors have done a nice job at revising the manuscript. Thank you.

Comments on the Quality of English Language

The English is satisfactory.